# FILLING IN THE GAP: ACHIEVING ROBUST AND ADAPTIVE GNNS THROUGH POST-PROCESSING

## ABSTRACT

Graph neural networks (GNNs) have shown significant success in modeling graph-structured data. However, their performance often deteriorates when faced with a change in the graph structure between training and test time, such as edge addition or removal—a common scenario considering the dynamic nature of graphs. To address this challenge, we propose FILLER (Framework for Integrating Layer-Level Edge-shift Recovery), a post-processing method which enhances the robustness of a GNN against edge sparsification while maintaining its adaptability to informative edge addition. Our key idea is to fill in the representation gap caused by edge distribution shift by injecting the *Edge-shift Recovery* (ER) layer into each layer of the GNN. Our ER layer is carefully designed to allow a GNN to maintain its high performance in dynamic graph environments even without any additional training, and its effectiveness is shown both theoretically and empirically. Our experiments on ten datasets for node classification and five GNN architectures demonstrate that FILLER is broadly applicable across diverse models and scenarios.

## 1 INTRODUCTION

Graph-structured data are commonly observed across various fields and applications. Since graphs capture crucial information about the relationships between entities, graph neural networks (GNNs) have proven particularly effective in semi-supervised learning, where observed data are insufficient to fully capture the relationship between features $X$ and labels $Y$ (Kipf & Welling, 2016; Hamilton et al., 2017; Veličković et al., 2017; Gasteiger et al., 2019; Brody et al., 2022). In these scenarios, the graph structure serves as a key manifold of high-dimensional features, significantly improving the estimation of the conditional distribution $p(Y|X)$ even with limited observations.

However, due to the dynamic nature of graphs, their structural information can change after training time, with edges being removed, sparsified, or new informative edges being added (Hu et al., 2020; Kazemi et al., 2020; Fu & He, 2022). Such changes can occur naturally in dynamic environments, but can also be introduced artificially for various purposes. For example, graphs are often sparsified to reduce the receptive field and enable faster inference (Ying et al., 2018; Chen et al., 2017; 2018). Conversely, edges are often added to give additional information to GNNs, enhancing performance (Chen et al., 2020; Alon & Yahav, 2020). As a consequence, it is common for a graph's connectivity to differ between training and testing time, a phenomenon referred to as *edge distribution shift*.

This shift is particularly problematic when edges are sparsified during inference, since it leads to a significant performance drop for most GNN models. The reduction in neighbor information limits the effectiveness of message propagation, which heavily depends on the graph's structure. Previous works (Hu et al., 2021; Yoo et al., 2019; Zhang et al., 2021b; Tian et al., 2022; Yang et al., 2024) have tackled this problem by using the graph structure only at training time and then discarding it at inference. Some approaches combine a multi-layer perceptron (MLP) with message propagation (Yoo et al., 2019), allowing the MLP to learn from the graph. Others distill the knowledge from a trained GNN into an MLP for similar purposes (Zhang et al., 2021b; Tian et al., 2022; Yang et al., 2024). These methods have demonstrated that structure-aware MLPs can match or even outperform GNNs, particularly when the test graph is sparsified due to edge distribution shift.

However, we argue that discarding edge information entirely during inference is not a fundamental solution for building models robust to edge distribution shift. Edges are not only removed but can also be added, introducing new, informative signals in dynamic graph environments. An optimally

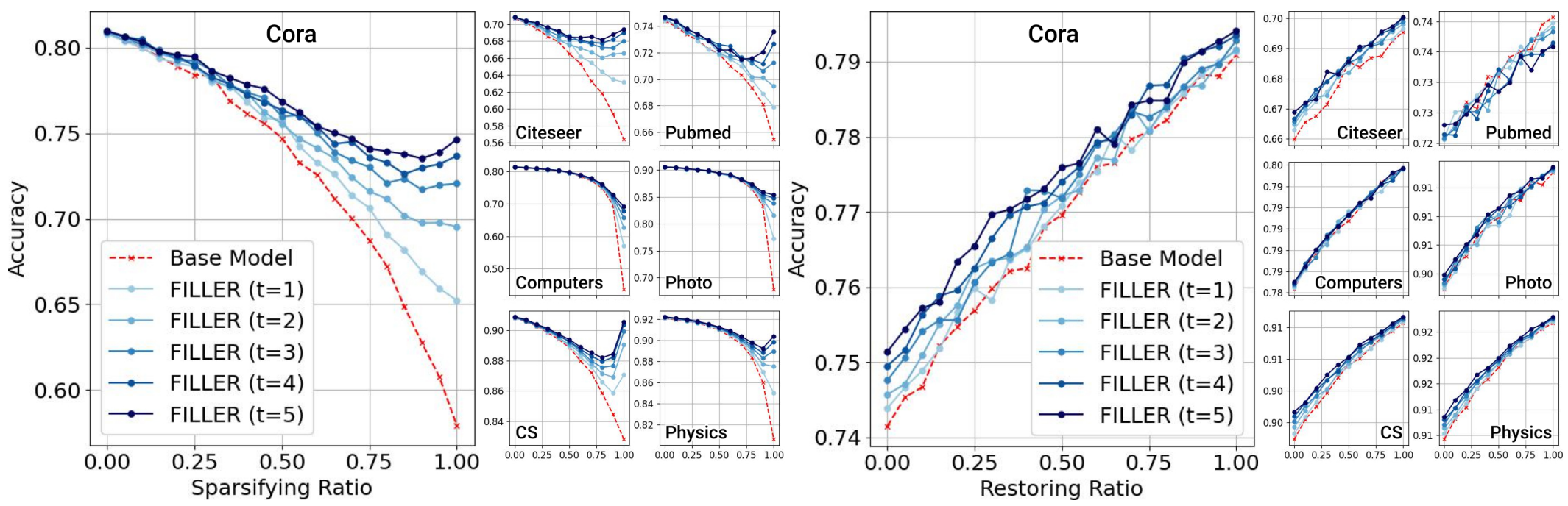

Figure 1: Performance change of SAGE with edge distribution shift: **(left)** gradual edge removal and **(right)** gradual addition of informative edges (i.e., edge restoration). The red line represents the base model, while progressively darker lines indicate the iterative application of our post-processing method, FILLER; it consistently improves the base performance on 7 different datasets.

robust model should be capable of leveraging these additional signals during test time to enhance its performance. None of the existing approaches fully address this crucial aspect of edge robustness: maintaining adaptability to newly added edges while ensuring robustness against edge removal.

In this work, we introduce FILLER (Framework for Integrating Layer-Level Edge-shift Recovery), a general post-processing method that can enhance any trained GNN to be robust against edge distribution shift. The key idea of FILLER is to inject an *Edge-shift Recovery* (ER) layer into each GNN layer, addressing the *representation gap* caused by the edge distribution shift, thereby restoring the model's original performance on test-time graphs. FILLER updates the ER layer through iterations, gradually improving the edge robustness of GNNs as illustrated in Fig 1. To the best of our knowledge, FILLER is the first approach that enhances the robustness of a trained GNN without requiring any additional training, while not discarding the graph structure given at inference.

We run extensive experiments on ten node classification benchmarks and demonstrate that FILLER consistently improves the robustness of GNNs across various types of datasets. Additionally, we apply FILLER to five well-known GNN architectures, showing its versatility and ability to be effectively integrated into existing GNN models. Our code is provided in the supplementary material.

## 2 PRELIMINARIES

**Notations.** Let $\mathcal{G} = (V, E)$ be an undirected graph, where $V$ is the set of nodes and $E$ is the set of edges. We denote the adjacency matrix of the graph by $\boldsymbol{A} \in \{0, 1\}^{|V| \times |V|}$, where $A_{ij} = 1$ if there is an edge between nodes $i$ and $j$, and $A_{ij} = 0$ otherwise. The feature matrix of the nodes is denoted by $\boldsymbol{X} \in \mathbb{R}^{|V| \times d_0}$, where $d_0$ is the dimensionality of node features.

**Graph Neural Networks.** Graph neural networks (GNNs) consist of multiple layers, where each layer performs two key operations: *aggregation* (AGG) and *update* (UPDATE) (Gilmer et al., 2017; Hu et al., 2019). The AGG operation gathers information from neighboring nodes, while the UPDATE operation combines this aggregated information with the node's previous representation. We denote the node representations after the $l$-th layer by $\boldsymbol{H}^{(l)} \in \mathbb{R}^{|V| \times d_l}$, where $d_l$ is its dimensionality. We also set $\boldsymbol{H}^{(0)} = \boldsymbol{X}$. Then, the $l$-th GNN layer is formally presented as

$$\boldsymbol{H}_{\mathcal{N}}^{(l)} = \mathrm{AGG}^{(l)}(\boldsymbol{H}^{(l-1)}, \boldsymbol{A}), \qquad \boldsymbol{H}^{(l)} = \mathrm{UPDATE}^{(l)}(\boldsymbol{H}_{\mathcal{N}}^{(l)}, \boldsymbol{H}^{(l-1)}).$$

Based on these AGG and UPDATE operations, we define a GNN $f$ with $L$ layers as a function of $\boldsymbol{X}$ and $\boldsymbol{A}$ parameterized with its layers: $\boldsymbol{H}^{(L)} = f(\boldsymbol{X}, \boldsymbol{A}; \{\mathrm{AGG}^{(l)}, \mathrm{UPDATE}^{(l)}\}_{l=1}^{L})$.

## 3 PROPOSED METHOD

We propose FILLER, a general post-processing method that can enhance the robustness of a trained GNN without requiring additional training. FILLER directly updates the GNN's message-passing

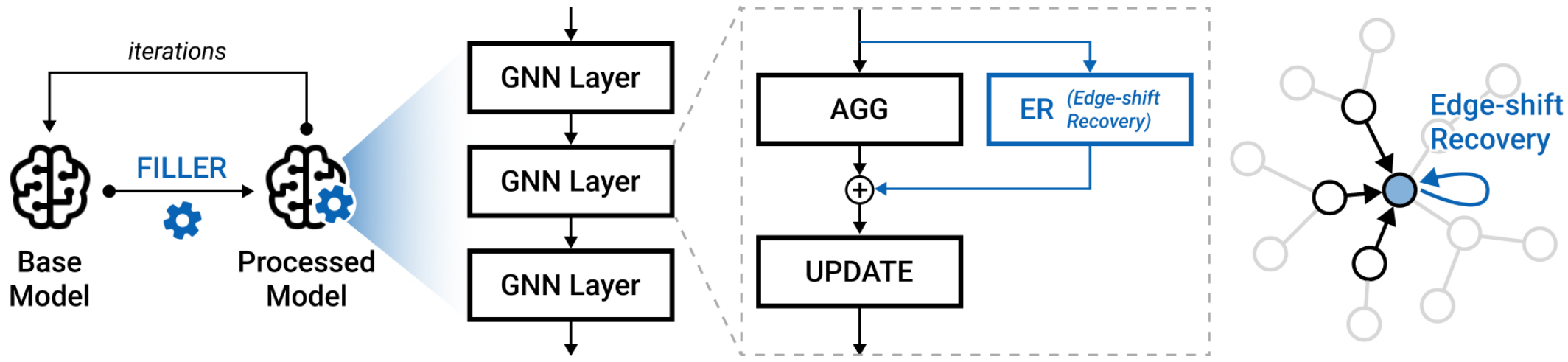

Figure 2: **(left)** FILLER enhances a trained GNN as a general post-processing method, **(middle)** by inserting our carefully-designed *Edge-shift Recovery* (ER) layer to each GNN layer, **(right)** which compensates for the altered signals that result from edge distribution shift at test time.

architecture, rather than creating a new model based on its learned knowledge, resulting in two key advantages: (a) our approach is broadly applicable to all GNNs that consist of AGG and UPDATE operations, and (b) the resulting model preserves its original characteristics when the graph remains intact while improving its performance when new, informative edges are added. A detailed algorithm that describes the overall process of FILLER is provided in Appendix A.

### 3.1 Overview of FILLER

Our key idea is to modify the GNN architecture in a layer-wise manner by introducing an *Edge-shift Recovery* (ER) layer as in Fig 2. The ER layer aims to recover for the perturbed edge connections that the model may encounter during inference. By addressing each layer individually, we can more effectively mitigate the impact of edge distribution shift, whose effect may vary across different layers. Specifically, our post-processing focuses on the AGG operation at each layer, since UPDATE is independent of the adjacency matrix. We formulate the layer-wise post-processing as follows:

$$\mathrm{AGG}_{t+1}^{(l)}(\boldsymbol{H}_t^{(l-1)}, \boldsymbol{A}) = \mathrm{AGG}_t^{(l)}(\boldsymbol{H}_t^{(l-1)}, \boldsymbol{A}) + k \cdot g(\boldsymbol{H}_t^{(l-1)}), \tag{1}$$

where $t$ is the number of post-processing iterations applied to the model and $k > 0$ is the step size which controls the magnitude of each update. Based on this framework, we aim to find a good ER layer $g$ that can effectively mitigate the performance degradation caused by edge shift.

**GAP.** To derive the objective of ER, we first define *edge distribution shift* as a test-time change of the adjacency matrix $\boldsymbol{A}$, which generates a perturbed adjacency matrix $\tilde{\boldsymbol{A}}$. Given $\tilde{\boldsymbol{A}}$ at test time, the $l$-th GNN layer produces perturbed node representations as follows:

$$\tilde{\boldsymbol{H}}_{\mathcal{N}}^{(l)} = \mathrm{AGG}^{(l)}(\boldsymbol{H}^{(l-1)}, \tilde{\boldsymbol{A}}), \qquad \tilde{\boldsymbol{H}}^{(l)} = \mathrm{UPDATE}^{(l)}(\tilde{\boldsymbol{H}}_{\mathcal{N}}^{(l)}, \boldsymbol{H}^{(l-1)}).$$

To address the performance degradation, at each layer $l$, the ER layer $g$ aims to restore the perturbed node representations to their original states given the output of layer $l-1$. Thus, $g$ is defined as a solution to the following problem, which we call *GNN Aggregation Perturbation* (GAP):

$$g(\boldsymbol{H}^{(l-1)}) = \arg\min_{g'} \|(\boldsymbol{H}_{\mathcal{N}}^{(l)} - \tilde{\boldsymbol{H}}_{\mathcal{N}}^{(l)}) - g'(\boldsymbol{H}^{(l-1)})\|_{\mathrm{F}}. \tag{2}$$

By focusing on the aggregated node representations before the UPDATE operation is applied, we leverage the fact that most activation functions used in GNNs are Lipschitz-continuous. This property ensures the difference in post-UPDATE representations is proportionally bounded by the difference in pre-UPDATE representations. Proofs of Lipschitz continuity for commonly used activation functions—ReLU, Sigmoid, and GELU—are provided in Appendix B.

**Essential Linearity of ER.** While the function $g$ could take any form, it is essential for our method that $g$ is a linear transformation: $g(\boldsymbol{H}^{(l-1)}) = \boldsymbol{H}^{(l-1)}\boldsymbol{W}^{(l)}$, for two key reasons. First, the transformation can be computed efficiently in parallel with the original AGG operation as shown in Fig. 2. As $g$ takes less time to calculate than AGG does, which already includes a linear transformation in most GNN architectures, the addition of $g$ does not increase the inference time. Second, as we apply our post-processing iteratively, the linearity of $g$ enables us to consolidate multiple linear transformations across iterations into a single weight matrix. This prevents the model size from growing as the number of post-processing iterations increases, ensuring full scalability.

### 3.2 Basic Version of Edge-Shift Recovery

**Surrogate Problem.** One challenge to solve GAP as in Eq. 2 is that $\tilde{\boldsymbol{H}}_{\mathcal{N}}^{(l)}$ is not observable during training time since edge distribution shift happens at test time. Thus, we propose *virtual perturbation* to simulate potential changes in the graph structure before testing. Assuming a random perturbation function $\phi$ which generalizes both edge removal and addition, we define virtual perturbation as the expected value of the perturbed node representations which can be calculated empirically:

$$\mathbb{E}_{\phi}[\tilde{\boldsymbol{H}}_{\mathcal{N}}^{(l)}] = \frac{1}{N}\sum\nolimits_{i=1}^{N} \mathrm{AGG}^{(l)}(\boldsymbol{H}^{(l-1)}, \phi(\boldsymbol{A})), \tag{3}$$

where $N$ is the number of trials to estimate the expectation.

**ER-Basic Layer.** With virtual perturbation, we propose the basic version of our *Edge-shift Recovery* (ER) layer as a solution to the solvable surrogate problem of GAP (GAP-surrogate) as follows:

$$g_B(\boldsymbol{H}^{(l-1)}) = \arg\min\nolimits_{g'} \|(\boldsymbol{H}_{\mathcal{N}}^{(l)} - \mathbb{E}_{\phi}[\tilde{\boldsymbol{H}}_{\mathcal{N}}^{(l)}]) - g'(\boldsymbol{H}^{(l-1)})\|_{\mathrm{F}}. \tag{4}$$

The choice of $\phi$ is a hyperparameter, with various algorithmic options available, including all types of graph structural augmentation (You et al., 2020; Zhang et al., 2021a; Li et al., 2023), sparsification (Rong et al., 2019; Chen et al., 2021), and partitioning (Chiang et al., 2019). In our framework, we make the simplest choice of $\phi$, setting $\phi(\mathbf{A}) = \mathbf{O}$ to simulate the most extreme case of edge removal which leaves no edges at test time. This approach is particularly efficient for virtual perturbation, as it eliminates the need for $N$ repetitive computations to estimate the expectation.

Since $g_B$ is a linear transformation, its weight matrix can be directly computed as follows:

$$\boldsymbol{W}^{(l)} = \boldsymbol{H}^{(l-1)+}(\boldsymbol{H}_{\mathcal{N}}^{(l)} - \mathbb{E}[\tilde{\boldsymbol{H}}_{\mathcal{N}}^{(l)}]), \tag{5}$$

where $+$ denotes the pseudoinverse of a matrix, which can be computed effectively through singular value decomposition (SVD). This method is more stable than training the weight matrix from scratch with an objective. Additional experiments comparing both methods are provided in Appendix C.

The time complexity of pseudoinverse is dominated by SVD, which is $O(mn \cdot \min(m, n))$ given a matrix of size $m \times n$ (Vasudevan & Ramakrishna, 2017). In our case, the target of pseudoinverse is $\boldsymbol{H}^{(l-1)} \in \mathbb{R}^{|V| \times d_{l-1}}$, and it simplifies the complexity into $O(|V| d_{l-1}^2)$ based on the safe assumption that $d_{l-1} \leq |V|$. This is the same complexity as that of a linear transformation if $d_{l-1} = d_l$, making our approach highly scalable to large graphs while not requiring backpropagation for gradient-based updates. Actual time consumption for pseudoinverse computation is provided in Appendix C.

### 3.3 Advanced Version of Edge-Shift Recovery

**Bridging GAP-surrogate to GAP.** As described in Eq. 2, an ideal ER should perform well across all types of perturbations. However, since $g_B$ is the solution to the GAP-surrogate problem, directly injecting $g_B$ into each GNN layer only guarantees improved performance when the test-time graph matches the virtual perturbation used during post-processing. We cannot be certain whether $g_B$ will result in a beneficial representation shift when the test graph differs from the virtually perturbed one. This is particularly important for practical use, as the processed model should maintain its original performance when the graph remains unchanged or undergoes minimal alteration.

To address this problem, we propose to generalize $g_B$ into $g_D = h(g_B(\boldsymbol{H}^{(l-1)}), \boldsymbol{A})$ by introducing a wrapper function $h$ which takes the adjacency matrix $\boldsymbol{A}$ as an additional input at test time. Its role is to adapt $g_B$ to a test graph unseen at both training and post-processing time, ensuring robustness for various perturbations. Let $\boldsymbol{\epsilon}^{(l)}(t_1, t_2, \boldsymbol{A}_1, \boldsymbol{A}_2) = \mathrm{AGG}_{t_1}^{(l)}(\boldsymbol{H}^{(l-1)}, \boldsymbol{A}_1) - \mathrm{AGG}_{t_2}^{(l)}(\boldsymbol{H}^{(l-1)}, \boldsymbol{A}_2)$ be the difference between the node representations at the $t_1$-th iteration with $\boldsymbol{A}_1$ and those at the $t_2$-th iteration with $\boldsymbol{A}_2$. We introduce three conditions $h$ should satisfy as follows:

- C1. **Recovery.** ER should fully compensate when the graph is perturbed equivalently to the virtual perturbation, which we set to $\boldsymbol{O}$ in our framework: $h(\boldsymbol{H}^{(l-1)}\boldsymbol{W}^{(l)}, \boldsymbol{O}) = \boldsymbol{H}^{(l-1)}\boldsymbol{W}^{(l)}$.
- C2. **Stability.** After each post-processing iteration, the new AGG function should be less deviated from the original AGG when there is no edge distribution shift compared to when such a shift occurs: $\|\boldsymbol{\epsilon}^{(l)}(t+1, t, \boldsymbol{A}, \boldsymbol{A})\|_{\mathrm{F}} < \|\boldsymbol{\epsilon}^{(l)}(t+1, t, \tilde{\boldsymbol{A}}, \tilde{\boldsymbol{A}})\|_{\mathrm{F}}$ for all $t$ and $l$.

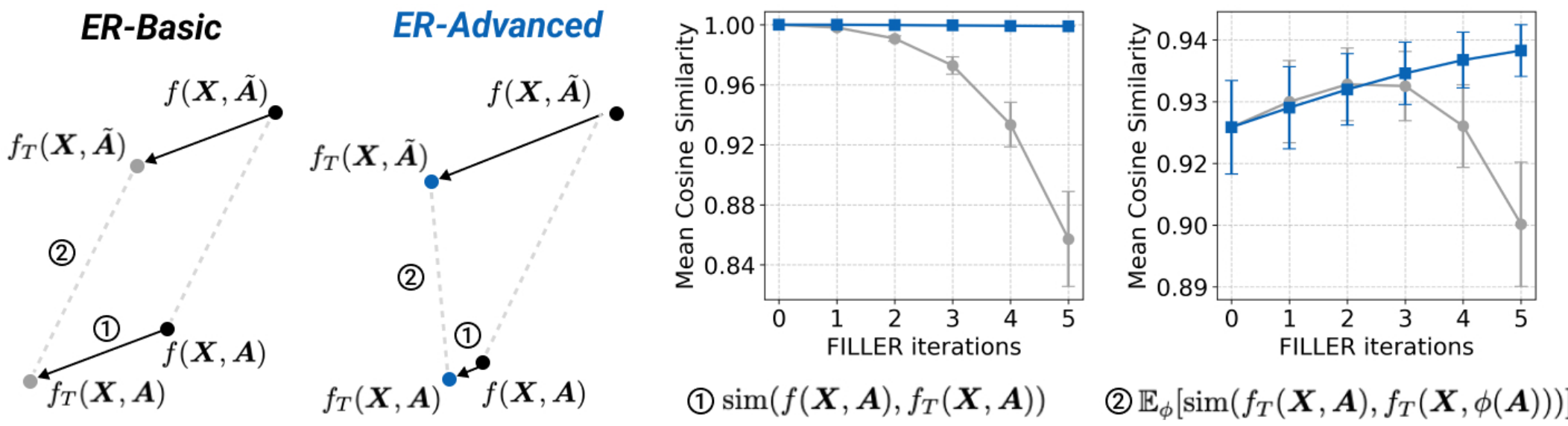

Figure 3: Illustration for a change in node representations before and after injecting ER. The right plots show the average cosine similarity between two representation matrices as the post-processing iterates, comparing ER-Basic (gray) and ER-Advanced (navy). ER-Advanced preserves the similarity in ① while increasing it in ②, demonstrating that is achieves C2 and C3 respectively.

C3. **Robustness.** The gap between node representations with and without perturbation, for any $\tilde{\boldsymbol{A}}$, decreases monotonically and eventually converges to zero as the number $t$ of post-processing iterations increases: $\delta_{t+1}^{(l)} < \delta_t^{(l)}$ and $\delta_\infty^{(l)} = 0$ for all $t$ and $l$ where $\delta_t^{(l)} = \|\boldsymbol{\epsilon}^{(l)}(t, t, \boldsymbol{A}, \tilde{\boldsymbol{A}})\|_\mathrm{F}$.

In this way, the GNN is optimized not only for the specific virtual perturbation, but also for various unseen perturbations including the original graph itself. While C1 and C2 are relatively straightforward to design with a satisfying function, C3 is challenging to implement and even harder to prove. Therefore, we propose a more intuitive condition that can ensure C3 (and even C2) as Lemma 3.1.

**Lemma 3.1.** *The following process-wise condition guarantees that both C2 and C3 hold:*

$$\forall l, \forall t, \quad \frac{\|\boldsymbol{\epsilon}^{(l)}(t+1, t, \tilde{\boldsymbol{A}}, \tilde{\boldsymbol{A}})\|_\mathrm{F}}{\|\boldsymbol{\epsilon}^{(l)}(t+1, t, \boldsymbol{A}, \boldsymbol{A})\|_\mathrm{F}} \geq 1 + \alpha,$$

*where $\tilde{\boldsymbol{A}}$ is a matrix obtained by masking some of the elements of $\boldsymbol{A}$ to zero, $\alpha > 0$ is any fixed constant, and $\|\cdot\|_\mathrm{F}$ represents the Frobenius norm.*

*Proof.* The proof is in Appendix D. □

**ER-Advanced Layer.** We propose $h_D$, the simplest and intuitive form of $h$ that satisfies all three conditions through multiplying a proper scaling coefficient to the transformation as follows:

$$h_D(g_B(\boldsymbol{H}^{(l-1)}), \boldsymbol{A}) = \left(\frac{|V|}{\sum_{i=1}^{|V|}\sum_{j=1}^{|V|}\hat{A}_{i,j}}\right)^n \cdot \boldsymbol{H}^{(l-1)}\boldsymbol{W}^{(l)},$$

where $\hat{\boldsymbol{A}} = \boldsymbol{A} + \boldsymbol{I}$ is the adjacency matrix with self-loops, and $n \geq 0$ is a hyperparameter which we set to 1 in all experiments. $h_D$ satisfies all three conditions above as stated in Theorem 3.2. Fig. 3 gives empirical evidence on how ER-Advanced achieves the proposed conditions. ER-Advanced maintains similarity in ① and gradually increases similarity in ②, demonstrating its stability and robustness, respectively. In contrast, ER-Basic begins to fail as $t$ increases.

**Theorem 3.2.** *$h_D$ satisfies all three conditions C1, C2, and C3 above.*

*Proof.* The proof is in Appendix D. □

## 4 Related Works

**Graph Distribution Shift.** Recent works have explored on addressing distribution shifts in graph learning, which occur when graph structures, node features, or labels change between training and inference (Wu et al., 2024). Graph domain adaptation aims to transfer knowledge from a training domain to a test domain (Xiao et al., 2024; Dai et al., 2022), while graph out-of-distribution learning focuses on generalizing models to unseen test graphs (Li et al., 2022). Graph continual learning addresses evolving graphs, where models aims to adapt to temporal changes while retaining previously learned information (Febrinanto et al., 2023). While these approaches typically focus on distribution shifts in specific scenarios, our work takes a different approach. In this work, we solve the problem focusing on edge distribution shift, where the edges can be removed or even added at test time.

**Using MLPs for Graph Tasks.** Recent works have considered multi-layer perceptrons (MLPs) as a scalable and robust alternative to graph neural networks (GNNs) for graph-based tasks. Unlike GNNs, MLPs do not rely on neighborhood information during inference, which makes them inherently robust to edge removal. BPN (Yoo et al., 2019) trains a structure-aware MLP by introducing a novel loss function that minimizes the difference between prior distributions and their post-belief propagation values. Graph-MLP (Hu et al., 2021) employs a contrastive loss to implicitly utilize the graph adjacency information during training, achieving effective performance even without explicit neighborhood information at test time. Other methods including NOSMOG (Tian et al., 2022) and GLNN (Zhang et al., 2021b) utilize knowledge distillation from GNNs to MLPs to transfer structural knowledge, achieving performance competitive with the teacher GNNs. VQGraph (Yang et al., 2024) extends this approach by learning discrete representations of local graph structures, enhancing MLP performance across various datasets. However, as MLPs ignore the neighborhood information at test graphs, they are unable to leverage new, informative edges given at test time; they lose edge adaptability, which is the core aspect of GNNs. In this work, we design a post-processing technique that can improve the robustness of GNNs while preserving their edge-adaptability.

# 5 EXPERIMENTS

**Datasets and Evaluation.** We measure node classification accuracy for seven small-scale datasets (Cora, Citeseer, Pubmed, Computers, Photo, CS, and Physics (Shchur et al., 2018)) and three large-scale datasets (Flickr (Zeng et al., 2019), Ogbn-arxiv (Hu et al., 2020), and Reddit (Hamilton et al., 2017)) in our experiments. For the small-scale datasets, we follow the data split configurations from previous work (Shchur et al., 2018) and repeat each scenario 10 times with random data splits and parameter initialization. For the large-scale datasets, we employ batch training with neighborhood sampling for scalability. We use the fixed public data splits due to the high variance in these datasets and run each experiment 5 times with random initialization.

**GNN Architectures.** We conduct experiments using five GNN architectures as base models: SAGE (Hamilton et al., 2017), GCN (Kipf & Welling, 2016), SGC (Wu et al., 2019), GAT (Veličković et al., 2017), and GIN (Xu et al., 2018), comparing their accuracy before and after applying FILLER. It is notable that we exclude GIN (Xu et al., 2018) from the large-scale experiments as it is not designed to work with neighbor sampling; GIN is originally designed to solve graph-level tasks such as graph classification. See Appendix E for how the $\mathrm{AGG}$ operation is defined in these GNN models.

**Hyperparameters.** For the small-scale datasets, we adopt GNNs with 2 layers and a hidden size of 64 (Shchur et al., 2018). For the large-scale datasets, we increase the model size to 3 layers with a hidden size of 256, which is a common practice for improved performance (Zeng et al., 2021). Other hyperparameters are kept as consistent as possible across the different architectures, as our primary objective is to evaluate performance gains from the post-processing rather than optimize individual models. The only exception is GIN (Xu et al., 2018), which requires different settings to train due to its unique design, as is not specifically designed for node classification tasks. Full details of the hyperparameters used in our experiments can be found in Appendix F.

## 5.1 MAIN EVALUATION

Our main experiments are designed to answer to the following three questions. We report accuracy on all GNN architectures for Q1, but focus on SAGE for Q2 and Q3 due to the lack of space. Refer to Appendix G for the full experimental results on different GNN architectures. Note that our results are consistent across the five different GNN architectures.

- Q1. Does FILLER preserve the original accuracy of GNNs when the graph is intact?
- Q2. Does FILLER improve the robustness of GNNs against edge removal?
- Q3. Does FILLER maintain the adaptability of GNNs to the addition of new edges?

**Q1. Preserving Original Accuracy.** We first show that FILLER preserves or often improves the accuracy of GNNs on the original graph when there is no edge distribution shift. This is particularly important for using FILLER in practice as a general post-processing method. In Table 1, we compare post-processed models with their original models when there is no edge distribution shift. We run paired t-tests for thorough comparison, and FILLER preserves the original accuracy in 42 out of 47

cases, demonstrating its practical stability. Additionally, in 12 cases, FILLER not only preserves but also improves the original performance at the 1% level. Later in ablation studies, we show that such a performance preservation is possible due to our well-designed ER-Advanced layer.

Table 1: Accuracy change after applying FILLER when there is no edge distribution shift. The bold indicates statistically significant at the 5% level ($p < 0.05$), while the color highlights the 1% level ($p < 0.01$), with red for improvements and blue for decrements.

| | Cora | Citeser | Pubemd | Computers | Photo | CS | Physics | Flickr | Ogbn-arxiv | Reddit |
|---|---|---|---|---|---|---|---|---|---|---|
| SAGE | 80.8 ± 2.0 | 70.7 ± 2.4 | 74.4 ± 2.4 | 81.4 ± 3.4 | 90.5 ± 2.7 | 90.8 ± 0.5 | 92.1 ± 1.3 | **53.1 ± 0.2** | **71.2 ± 0.2** | 96.3 ± 0.0 |
| +FILLER | 81.0 ± 2.0 | 70.8 ± 2.2 | **74.7 ± 2.2** | 81.4 ± 3.3 | 90.5 ± 2.7 | **90.9 ± 0.4** | **92.2 ± 1.3** | 52.9 ± 0.2 | 71.1 ± 0.2 | 96.3 ± 0.0 |
| Δ | ↑ 0.2% | ↑ 0.2% | ↑ 0.4% | ↑ 0.1% | ↑ 0.0% | ↑ 0.1% | ↑ 0.1% | ↓ 0.3% | ↓ 0.2% | ↓ 0.0% |
| p-value | 0.0975 | 0.1519 | 0.0432 | 0.0655 | 0.3880 | 0.0053 | 0.0058 | 0.0002 | 0.0014 | 0.7780 |
| GCN | 82.4 ± 1.3 | 71.0 ± 1.9 | 77.5 ± 1.6 | 83.5 ± 2.3 | 91.3 ± 1.5 | 91.0 ± 0.4 | 92.7 ± 0.9 | **52.7 ± 0.2** | 71.3 ± 0.1 | **94.0 ± 0.0** |
| +FILLER | **82.8 ± 1.4** | **71.5 ± 1.9** | 77.7 ± 1.6 | 83.6 ± 2.3 | 91.3 ± 1.5 | 91.1 ± 0.3 | **92.8 ± 0.9** | 52.0 ± 0.2 | 71.2 ± 0.1 | 94.0 ± 0.0 |
| Δ | ↑ 0.4% | ↑ 0.7% | ↑ 0.2% | ↑ 0.0% | ↓ 0.0% | ↑ 0.0% | ↑ 0.1% | ↓ 1.3% | ↓ 0.1% | ↓ 0.0% |
| p-value | 0.0280 | 0.0012 | 0.2618 | 0.4586 | 0.8504 | 0.0638 | 0.0003 | 0.0033 | 0.1150 | 0.0205 |
| SGC | 82.0 ± 1.7 | 69.8 ± 1.8 | 75.8 ± 2.3 | 83.3 ± 1.5 | 91.0 ± 1.8 | 90.9 ± 0.6 | 92.7 ± 1.3 | **51.5 ± 0.1** | **69.1 ± 0.1** | 94.2 ± 0.0 |
| +FILLER | **82.3 ± 1.5** | **70.1 ± 1.9** | 75.9 ± 2.4 | 83.3 ± 1.5 | 91.0 ± 1.8 | **91.0 ± 0.6** | **92.8 ± 1.3** | 51.0 ± 0.1 | 69.1 ± 0.1 | 94.1 ± 0.0 |
| Δ | ↑ 0.4% | ↑ 0.4% | ↑ 0.2% | ↑ 0.0% | ↑ 0.0% | ↑ 0.1% | ↑ 0.1% | ↓ 0.9% | ↓ 0.1% | ↓ 0.0% |
| p-value | 0.0077 | 0.0320 | 0.1879 | 0.3321 | 0.4090 | 0.0035 | 0.0011 | 0.0012 | 0.0391 | 0.1708 |
| GAT | 82.0 ± 1.6 | 70.9 ± 1.5 | 77.4 ± 1.8 | 84.3 ± 2.1 | 91.5 ± 1.3 | 89.5 ± 0.4 | 91.6 ± 1.6 | **54.1 ± 0.2** | **71.5 ± 0.2** | 94.0 ± 0.1 |
| +FILLER | 82.3 ± 1.5 | 71.0 ± 1.5 | **78.0 ± 1.7** | 84.4 ± 2.0 | 91.5 ± 1.2 | 89.5 ± 0.4 | **91.7 ± 1.5** | 53.5 ± 0.1 | 71.4 ± 0.1 | 94.0 ± 0.1 |
| Δ | ↑ 0.3% | ↑ 0.1% | ↑ 0.7% | ↑ 0.1% | ↓ 0.0% | ↓ 0.0% | ↑ 0.1% | ↓ 1.3% | ↓ 0.2% | ↓ 0.0% |
| p-value | 0.0914 | 0.6248 | 0.0058 | 0.2167 | 0.7192 | 0.7028 | 0.0038 | 0.0008 | 0.0350 | 0.2635 |
| GIN | 79.5 ± 2.3 | 67.7 ± 2.7 | 75.3 ± 3.2 | 76.9 ± 3.1 | 87.2 ± 2.1 | 85.3 ± 1.1 | 89.9 ± 1.5 | - | - | - |
| +FILLER | 79.5 ± 2.0 | 67.4 ± 2.7 | 75.8 ± 3.3 | **77.0 ± 3.1** | **87.5 ± 2.1** | **85.8 ± 1.1** | **90.2 ± 1.4** | - | - | - |
| Δ | ↓ 0.0% | ↓ 0.4% | ↑ 0.7% | ↑ 0.1% | ↑ 0.3% | ↑ 0.7% | ↑ 0.3% | - | - | - |
| p-value | 0.9181 | 0.3165 | 0.1691 | 0.0300 | 0.0000 | 0.0000 | 0.0001 | - | - | - |

**Q2. Robustness Against Edge Removal.** To evaluate how well FILLER improves the robustness of GNNs against edge removal, we run experiments by gradually removing edges from the original graphs. We train each GNN on the original graph with all edges, and drop 50%, 75%, and 100% of the edges randomly at test time. FILLER is applied right after the training of GNNs.

As shown in Table 2, FILLER significantly enhances the performance of SAGE on sparsified graphs across all datasets. On average across all datasets, FILLER improves performance by 0.70%, 2.66%, and 21.5% when 50%, 75%, and 100% of the edges are removed, respectively. Although we report detailed results only for SAGE to save space, FILLER shows consistent improvement over the base models across various types of GNN architectures as presented in Appendix G.

Table 2: Accuracy after applying FILLER with 50%, 75%, and 100% removal of edges. The numbers are in bold and colored in the same manner as in Table 1. FILLER consistently and significantly improves the accuracy of the base model, showing its effectiveness for improving robustness.

| | -50% | | | -75% | | | -100% | | |
|---|---|---|---|---|---|---|---|---|---|
| **Dataset** | SAGE | + FILLER | Δ | SAGE | + FILLER | Δ | SAGE | + FILLER | Δ |
| Cora | 74.7 ± 2.6 | **76.9 ± 2.0** | ↑ 2.9% | 68.7 ± 3.5 | **74.1 ± 2.3** | ↑ 7.8% | 57.9 ± 4.9 | **74.6 ± 2.1** | ↑ 28.9% |
| Citeseer | 66.5 ± 2.4 | **68.5 ± 2.0** | ↑ 3.0% | 62.7 ± 2.7 | **68.5 ± 2.2** | ↑ 9.3% | 56.4 ± 4.5 | **69.4 ± 2.5** | ↑ 23.1% |
| Pubmed | 71.9 ± 4.0 | 72.2 ± 4.2 | ↑ 0.4% | 69.5 ± 5.5 | **71.7 ± 5.6** | ↑ 3.2% | 65.4 ± 8.8 | **73.6 ± 3.9** | ↑ 12.5% |
| Computers | 79.6 ± 3.3 | 79.8 ± 3.3 | ↑ 0.2% | 76.3 ± 3.6 | **77.2 ± 3.2** | ↑ 1.2% | 43.5 ± 10.1 | **69.2 ± 4.8** | ↑ 58.9% |
| Photo | 89.4 ± 2.8 | 89.4 ± 2.7 | ↑ 0.0% | 87.3 ± 2.8 | **87.8 ± 2.8** | ↑ 0.6% | 67.8 ± 7.5 | **85.4 ± 2.4** | ↑ 25.9% |
| CS | 88.8 ± 0.5 | **89.4 ± 0.5** | ↑ 0.6% | 86.5 ± 0.8 | **88.2 ± 0.5** | ↑ 2.0% | 82.8 ± 2.4 | **90.6 ± 0.5** | ↑ 9.3% |
| Physics | 91.0 ± 1.5 | **91.3 ± 1.4** | ↑ 0.3% | 89.0 ± 2.0 | **90.1 ± 1.6** | ↑ 1.2% | 80.6 ± 5.7 | **90.4 ± 1.6** | ↑ 12.1% |
| Flickr | 50.2 ± 0.2 | 50.1 ± 0.2 | ↓ 0.4% | 46.2 ± 0.6 | 46.4 ± 0.5 | ↑ 0.5% | 41.9 ± 0.5 | **46.2 ± 0.3** | ↑ 10.2% |
| Ogbn-arxiv | 67.2 ± 0.2 | 67.1 ± 0.2 | ↓ 0.1% | 61.7 ± 0.3 | **62.3 ± 0.1** | ↑ 0.9% | 44.3 ± 1.1 | **49.8 ± 1.0** | ↑ 12.6% |
| Reddit | 95.7 ± 0.0 | 95.7 ± 0.0 | ↑ 0.0% | 94.9 ± 0.1 | 94.9 ± 0.1 | ↑ 0.0% | 45.9 ± 1.1 | **55.9 ± 1.2** | ↑ 21.9% |

**Q3. Adaptability to Edge Addition.** Edge distribution shift is not limited to edge removal; edges can also be added to the test graph. However, simulating edge addition through a random function, as done with edge removal, introduces noisy edges that do not contribute meaningful information to solving the task. Therefore, we propose an *edge restoration* scenario, where a GNN is trained on a graph with a predefined fraction of edges removed and tested as these removed edges are gradually restored. In this way, the added edges are informative since they come from the original graph. The goal of this experiment is to evaluate the model's adaptability to new, informative edges.

We present the results in Table 3, where GNNs are trained on 60% of the edges and tested with 0%, 50%, and 100% of the excluded edges restored. The post-processed models improve their accuracy as the base models do as more informative edges are restored, demonstrating that FILLER preserves

the edge-adaptability of GNNs. Specifically, the post-processed SAGE do not show accuracy drop at the 1% significance level in all 30 cases, while showing an improvement in 9 cases.

Table 3: Accuracy change after applying FILLER with 0%, 50%, and 100% restoration of missing edges. The numbers are in bold and colored in the same way as in Table 1. FILLER does not lose the adaptability of GNNs to informative edges even after improving their robustness.

| | **0%** | | | **+50%** | | | **+100%** | | |
|---|---|---|---|---|---|---|---|---|---|
| **Dataset** | SAGE | + FILLER | Δ | SAGE | + FILLER | Δ | SAGE | + FILLER | Δ |
| Cora | $74.2 \pm 2.3$ | **$75.1 \pm 2.0$** | ↑ 1.3% | $77.0 \pm 2.0$ | **$77.6 \pm 2.1$** | ↑ 0.8% | $79.1 \pm 1.9$ | **$79.4 \pm 2.0$** | ↑ 0.4% |
| Citeseer | $66.0 \pm 3.1$ | **$66.9 \pm 3.3$** | ↑ 1.4% | $68.6 \pm 2.8$ | $68.6 \pm 2.9$ | ↑ 0.0% | $69.6 \pm 2.3$ | **$70.0 \pm 2.3$** | ↑ 0.7% |
| Pubmed | $72.6 \pm 2.6$ | $72.8 \pm 3.0$ | ↑ 0.3% | $73.6 \pm 2.8$ | $73.4 \pm 2.8$ | ↓ 0.3% | $74.6 \pm 2.7$ | $74.1 \pm 2.9$ | ↓ 0.6% |
| Computers | $78.4 \pm 2.9$ | $78.5 \pm 2.9$ | ↑ 0.1% | $79.1 \pm 3.1$ | $79.1 \pm 3.0$ | ↑ 0.1% | $79.6 \pm 3.1$ | $79.6 \pm 3.1$ | ↑ 0.0% |
| Photo | $90.3 \pm 2.3$ | **$90.4 \pm 2.3$** | ↑ 0.1% | $90.8 \pm 2.2$ | $90.9 \pm 2.2$ | ↑ 0.1% | $91.1 \pm 2.1$ | $91.1 \pm 2.1$ | ↑ 0.0% |
| CS | $89.2 \pm 0.4$ | **$89.7 \pm 0.4$** | ↑ 0.5% | $90.4 \pm 0.5$ | **$90.5 \pm 0.5$** | ↑ 0.1% | $91.1 \pm 0.5$ | **$91.2 \pm 0.5$** | ↑ 0.1% |
| Physics | $91.2 \pm 0.8$ | **$91.3 \pm 0.9$** | ↑ 0.2% | $91.7 \pm 0.9$ | $91.8 \pm 1.0$ | ↑ 0.1% | $92.1 \pm 1.0$ | $92.1 \pm 1.0$ | ↑ 0.1% |
| Flickr | $50.9 \pm 0.2$ | $50.9 \pm 0.1$ | ↓ 0.0% | **$51.6 \pm 0.2$** | $51.5 \pm 0.2$ | ↓ 0.3% | $52.0 \pm 0.3$ | $51.9 \pm 0.3$ | ↓ 0.1% |
| Ogbn-arxiv | $69.0 \pm 0.0$ | $68.9 \pm 0.1$ | ↓ 0.1% | **$70.2 \pm 0.1$** | $70.1 \pm 0.0$ | ↓ 0.2% | **$71.0 \pm 0.2$** | $71.0 \pm 0.1$ | ↓ 0.1% |
| Reddit | $95.9 \pm 0.1$ | $95.9 \pm 0.1$ | ↓ 0.0% | $96.1 \pm 0.0$ | **$96.1 \pm 0.0$** | ↑ 0.0% | $96.3 \pm 0.0$ | $96.3 \pm 0.0$ | ↓ 0.0% |

### 5.2 Ablation Studies

Our ablation studies consist of various experiments to provide a better understanding of FILLER for the following perspectives, going beyond typical hyperparameter studies:

Q4. How much better is ER-Advanced compared to ER-Basic?

Q5. How does the random perturbation function $\phi(\boldsymbol{A})$ affect FILLER?

Q6. Does FILLER work across different GNN hyperparameters?

Q7. Is it better to create ER layers through pseudoinverse than gradient-based training?

**Q4. Effectiveness of ER-Advanced.** In Fig. 4, we evaluate the effectiveness of the ER-Advanced layer compared to ER-Basic through experiments. As anticipated, ER-Basic struggles to maintain performance on the original graph (related to C2) and shows inconsistent results across varying levels of edge removal (related to C3), while it performs well when the edge removal ratio is 1 (related to C1) since this case aligns with the virtual perturbation. In contrast, ER-Advanced demonstrates consistent, robust performance improvements, validating the effectiveness of its design.

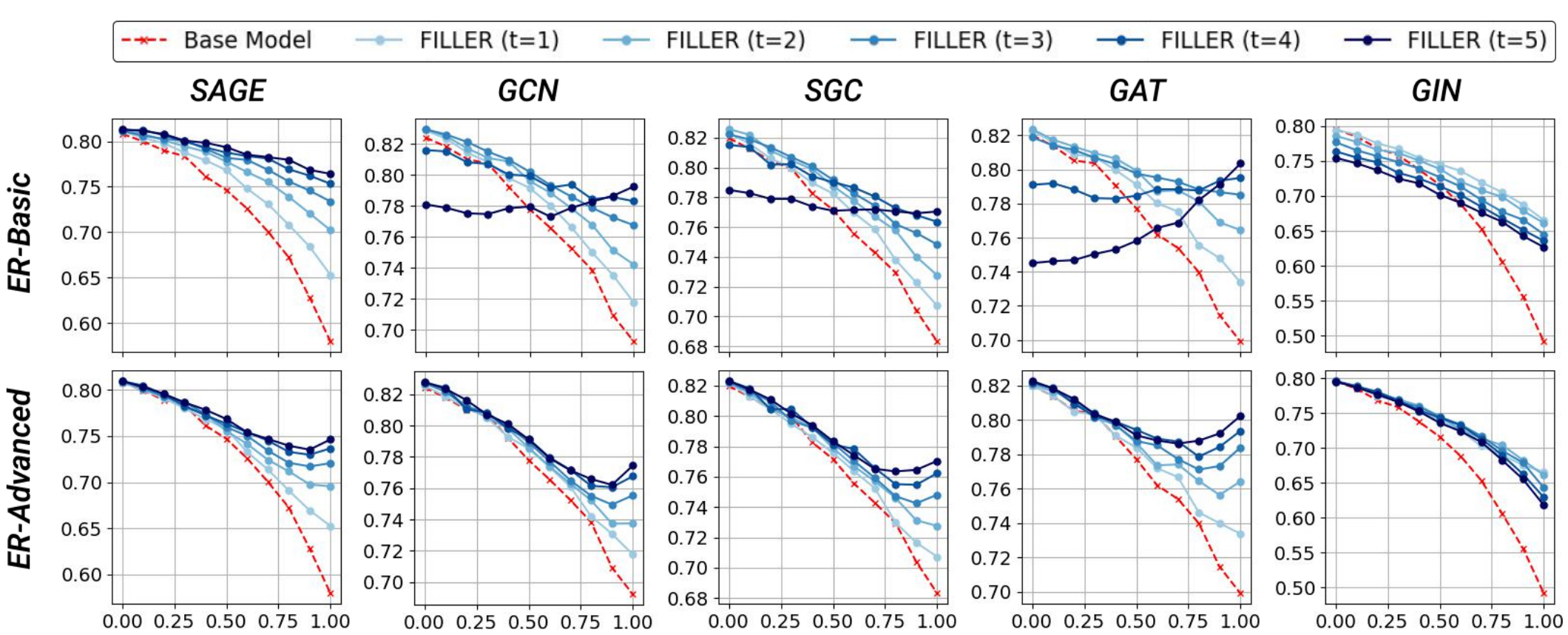

Figure 4: Comparison between ER-Advanced and ER-Basic. ER-Advanced **(bottom)** consistently improves performance on various perturbed test graphs, while ER-Basic **(top)** does not.

**Q5. Non-deterministic Virtual Perturbation.** In our experiments, we used full edge removal as the simplest deterministic perturbation function $\phi$ for virtual perturbation, though it can be replaced with other alternatives. Here we test the random edge removal function as $\phi_p(\boldsymbol{A})$, which removes a fraction $p$ of edges from the graph non-deterministically. We run $\phi$ for 10 times at each iteration to

compute the expected value as shown in Eq. 3. Then, to meet our first condition (C1) of the wrapper function in ER-Advanced, we define a new wrapper function $h_R$ as follows:

$$h_R(g_B(\boldsymbol{H}^{(l-1)}), \boldsymbol{A}) = \left( \frac{2(1-p) \cdot |E_{train}| + |V|}{\sum_{i=1}^{|V|} \sum_{j=1}^{|V|} \hat{A}_{i,j}} \right)^n \cdot \boldsymbol{H}^{(l-1)} \boldsymbol{W}^{(l)},$$

where $p$ is set to 0.25, 0.5, 0.75, or 1.0, and $|E_{\text{train}}|$ is the number of edges in the training graph. We skip to prove how $h_R$ satisfies all three conditions C1, C2, and C3 since it is straightforward.

As shown in Table 4, the post-processed models consistently improve the base models regardless of which perturbation function is used. Nevertheless, our simplest version $\phi(\boldsymbol{A}) = \boldsymbol{O}$ works generally well, while being the fastest due to its deterministic nature. This is possibly because the model can be post-processed expecting the most extreme edge perturbation in a stable way. Thus, we recommend using the deterministic perturbation function as the default option for practical use.

Table 4: Accuracy change of SAGE after applying FILLER with random perturbation functions $\phi_p$. FILLER improves the base model in all choices, while our deterministic version works best.

| | Cora | | | Citeseer | | | Pubmed | | |
|---|---|---|---|---|---|---|---|---|---|
| | −50% | −75% | −100% | −50% | −75% | −100% | −50% | −75% | −100% |
| Base | $74.7 \pm 2.6$ | $68.7 \pm 3.5$ | $57.9 \pm 4.9$ | $66.5 \pm 2.4$ | $62.7 \pm 2.7$ | $56.4 \pm 4.5$ | $71.9 \pm 4.0$ | $69.5 \pm 5.5$ | $65.4 \pm 8.8$ |
| $\phi_{0.25}$ | $75.2 \pm 1.9$ | $69.8 \pm 3.0$ | $61.9 \pm 3.9$ | $67.3 \pm 1.9$ | $65.2 \pm 2.5$ | $61.5 \pm 3.4$ | $72.0 \pm 4.1$ | $71.1 \pm 5.3$ | $70.9 \pm 7.1$ |
| $\phi_{0.5}$ | $76.0 \pm 2.0$ | $71.7 \pm 2.8$ | $66.9 \pm 2.8$ | $68.0 \pm 1.8$ | $66.6 \pm 2.7$ | $64.5 \pm 3.4$ | $72.1 \pm 4.2$ | $72.1 \pm 5.5$ | $73.8 \pm 3.3$ |
| $\phi_{0.75}$ | $76.7 \pm 1.9$ | $73.5 \pm 2.3$ | $72.0 \pm 1.9$ | $\mathbf{68.5 \pm 1.9}$ | $67.8 \pm 2.4$ | $67.5 \pm 2.4$ | $\mathbf{72.3 \pm 4.2}$ | $\mathbf{72.2 \pm 5.4}$ | $\mathbf{74.2 \pm 3.1}$ |
| $\phi_{1.0}$ (ours) | $\mathbf{76.9 \pm 2.0}$ | $\mathbf{74.1 \pm 2.3}$ | $\mathbf{74.6 \pm 2.1}$ | $68.5 \pm 2.0$ | $\mathbf{68.5 \pm 2.2}$ | $\mathbf{69.4 \pm 2.5}$ | $72.2 \pm 4.2$ | $71.7 \pm 5.6$ | $73.6 \pm 3.9$ |

**Q6. GNN Architectural Hyperparameters.** As a general post-processing framework, we expect FILLER to succeed not only for various GNN models, but also for various choices of their architectural hyperparameters. In Figure 5, we run experiments by varying key hyperparameters of SAGE, including the number of layers, hidden dimension sizes, and activation functions. FILLER enhances the performance consistently across all tested configurations, even when the base model shows poor accuracy as it has too many layers or too small hidden size. This indicates that FILLER is broadly applicable to various GNN models regardless of their hyperparameter configurations.

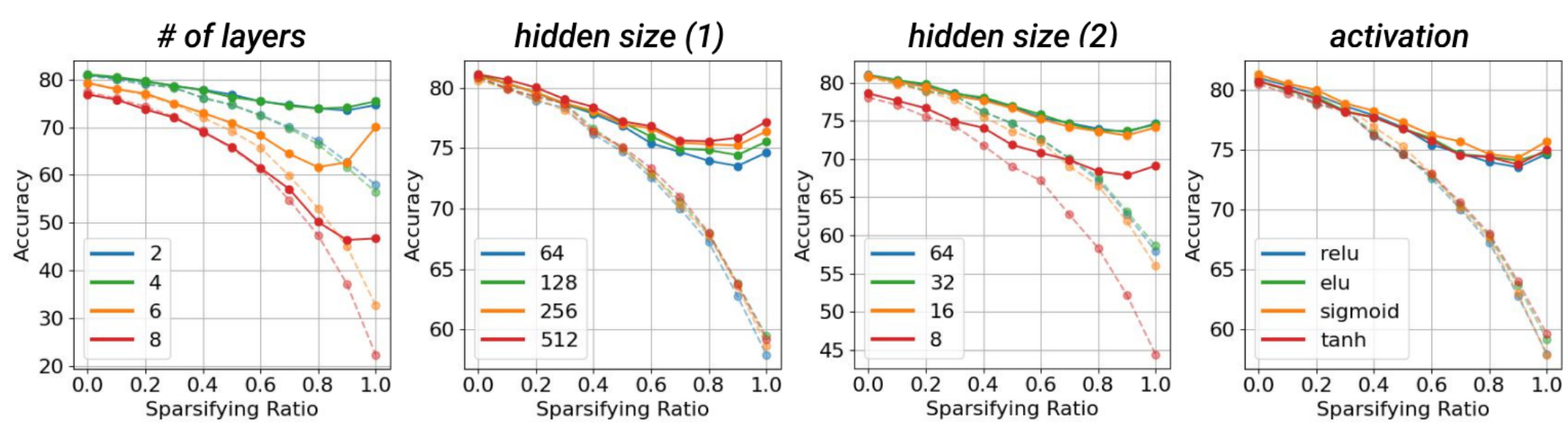

Figure 5: The effectiveness of FILLER for edge removal at Cora with various architectural hyperparameters of SAGE. The dashed lines represent the base models, while the solid lines represent the post-processed models. FILLER consistently improves the base accuracy in all cases.

**Q7. Advantage of Psuedoinverse.** We compare our pseudoinverse-based way to get $\boldsymbol{W}^{(l)}$ in ER layers with the gradient-based training. Detailed results are in Appendix C. We find that our method is more effective than the gradient-based approach, while being faster especially in large graphs.

## 6 Conclusion

In this work, we introduce FILLER (Framework for Integrating Layer-Level Edge-shift Recovery), a general post-processing method for enhancing the robustness and adaptability of GNN models in the face of edge distribution shift during inference. By inserting the *Edge-shift Recovery* (ER) module into each GNN layer, FILLER effectively mitigates performance degradation while capitalizing on newly introduced, informative edges. The effectiveness of our approach is theoretically justified in terms of key characteristics–recovery, stability, and robustness, and empirically demonstrated on ten node classification benchmarks and five different GNN architectures.

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

# A ALGORITHMIC EXPLANATION OF FILLER

FILLER can be broken down into three key steps. First, as in Algorithm 1, node representations extraction is performed during a single inference pass on the training graph.

**Algorithm 1** Constructing Layer-wise and Aggregated Node Representations

1: **Input:** Graph $\mathcal{G} = (V, E)$ with node features $\boldsymbol{X}$, adjacency matrix $\boldsymbol{A}$, and a trained GNN with $L$ layers.
2: **Output:** Layer-wise node representations $\{\boldsymbol{H}^{(0)}, \boldsymbol{H}^{(1)}, \ldots, \boldsymbol{H}^{(L)}\}$ and aggregated representations $\{\boldsymbol{H}_{\mathcal{N}}^{(1)}, \boldsymbol{H}_{\mathcal{N}}^{(2)}, \ldots, \boldsymbol{H}_{\mathcal{N}}^{(L)}\}$.
3: Initialize input features: $\boldsymbol{H}^{(0)} \leftarrow \boldsymbol{X}$
4: Initialize set of layer-wise representations: $\mathcal{H} \leftarrow \{\boldsymbol{H}^{(0)}\}$
5: Initialize set of aggregated representations: $\mathcal{H}_{\mathcal{N}} \leftarrow \{\}$
6: **for** each layer $l = 1$ to $L$ **do**
7: Compute aggregated representation and add to aggregated set:

$$\boldsymbol{H}_{\mathcal{N}}^{(l)} \leftarrow \mathrm{AGG}^{(l)}(\boldsymbol{H}^{(l-1)}, \boldsymbol{A}), \qquad \mathcal{H}_{\mathcal{N}} \leftarrow \mathcal{H}_{\mathcal{N}} \cup \{\boldsymbol{H}_{\mathcal{N}}^{(l)}\}$$

8: Update node representation at layer $l$ and add to layer-wuse set:

$$\boldsymbol{H}^{(l)} \leftarrow \mathrm{UPDATE}^{(l)}(\boldsymbol{H}_{\mathcal{N}}^{(l)}, \boldsymbol{H}^{(l-1)}), \qquad \mathcal{H} \leftarrow \mathcal{H} \cup \{\boldsymbol{H}^{(l)}\}$$

9: **end for**
10: **Return:** Layer-wise representations $\mathcal{H}$ and aggregated representations $\mathcal{H}_{\mathcal{N}}$.

Next, in Algorithm 2, the weight matrix for the ER layer is computed. This involves calculating the virtual perturbation, pseudoinverse, and performing matrix subtraction and multiplication at each layer. Since we use a deterministic approach for virtual perturbation, this part can be completed efficiently with a single AGG operation.

**Algorithm 2** Calculating Weight Matrices for ER Layer at Each Layer

1: **Input:** Layer-wise node representations $\mathcal{H} = \{\boldsymbol{H}^{(0)}, \boldsymbol{H}^{(1)}, \ldots, \boldsymbol{H}^{(L)}\}$, aggregated representations $\mathcal{H}_{\mathcal{N}} = \{\boldsymbol{H}_{\mathcal{N}}^{(1)}, \boldsymbol{H}_{\mathcal{N}}^{(2)}, \ldots, \boldsymbol{H}_{\mathcal{N}}^{(L)}\}$, adjacency matrix $\boldsymbol{A}$, random perturbation function $\phi$ and a trained GNN with $L$ layers.
2: **Output:** Set of weight matrices $\{\boldsymbol{W}^{(1)}, \ldots, \boldsymbol{W}^{(L)}\}$ for the ER layer at each layer.
3: Initialize set of weight matrices: $\mathcal{W} \leftarrow \{\}$
4: **for** each layer $l = 1$ to $L$ **do**
5: Calculate virtual perturbation empirically:

$$\mathbb{E}_{\phi}[\tilde{\boldsymbol{H}}_{\mathcal{N}}^{(l)}] = \frac{1}{N}\sum_{i=1}^{N} \mathrm{AGG}^{(l)}(\boldsymbol{H}^{(l-1)}, \phi(\boldsymbol{A}))$$

6: Compute pseudoinverse of node representations at previous layer: $\boldsymbol{H}^{(l-1)+}$
7: Compute weight matrix for layer $l$ and add to weight set:

$$\boldsymbol{W}^{(l)} = \boldsymbol{H}^{(l-1)+}(\boldsymbol{H}_{\mathcal{N}}^{(l)} - \mathbb{E}[\tilde{\boldsymbol{H}}_{\mathcal{N}}^{(l)}]), \qquad \mathcal{W} \leftarrow \mathcal{W} \cup \{\boldsymbol{W}^{(l)}\}$$

8: **end for**
9: **Return:** Set of weight matrices $\mathcal{W}$

Finally, FILLER updates the architecture of the trained GNN using the set of obtained weight matrices $\mathcal{W} = \{\boldsymbol{W}^{(1)}, \ldots, \boldsymbol{W}^{(L)}\}$ and an appropriately designed wrapper function $h_D$, integrating the ER layer in parallel with the AGG operation.

$$\mathrm{AGG}_{t+1}^{(l)}(\boldsymbol{H}_t^{(l-1)}, \boldsymbol{A}) = \mathrm{AGG}_t^{(l)}(\boldsymbol{H}_t^{(l-1)}, \boldsymbol{A}) + k \cdot h_D(\boldsymbol{H}^{(l-1)}\boldsymbol{W}^{(l)})$$

# B Lipschitz Continuity

Activation functions play a pivotal role in Graph Neural Networks (GNNs) by introducing non-linearity, which enables the network to model complex relationships within graph-structured data. Ensuring that these activation functions are Lipschitz continuous is essential for guaranteeing that similarly aggregated representation can result a simliar output after applying the activation function. In this section, we formally derive the Lipschitz continuity of three widely used activation functions: Rectified Linear Unit (ReLU), Sigmoid, and Gaussian Error Linear Unit(GELU).

## B.1 Definition of Lipschitz Continuity

A function $f : \mathbb{R} \to \mathbb{R}$ is said to be **Lipschitz continuous** if there exists a constant $L \geq 0$ such that for all $x, y \in \mathbb{R}$,

$$|f(x) - f(y)| \leq L|x - y|.$$

## B.2 Rectified Linear Unit (ReLU)

The Rectified Linear Unit (ReLU) activation function is defined as:

$$\text{ReLU}(x) = \max(0, x).$$

To prove that ReLU is 1-Lipschitz continuous, we need to show that:

$$|\text{ReLU}(x) - \text{ReLU}(y)| \leq |x - y| \quad \forall x, y \in \mathbb{R}.$$

**Case 1:** $x \geq 0$ and $y \geq 0$

In this case,

$$\text{ReLU}(x) = x \quad \text{and} \quad \text{ReLU}(y) = y.$$

Thus,

$$|\text{ReLU}(x) - \text{ReLU}(y)| = |x - y| \leq |x - y|.$$

**Case 2:** $x < 0$ and $y < 0$

Here,

$$\text{ReLU}(x) = 0 \quad \text{and} \quad \text{ReLU}(y) = 0.$$

Therefore,

$$|\text{ReLU}(x) - \text{ReLU}(y)| = |0 - 0| = 0 \leq |x - y|.$$

**Case 3:** $x \geq 0$ and $y < 0$ (without loss of generality)

In this scenario,

$$\text{ReLU}(x) = x \quad \text{and} \quad \text{ReLU}(y) = 0.$$

Thus,

$$|\text{ReLU}(x) - \text{ReLU}(y)| = |x - 0| = |x| \leq |x - y|.$$

This inequality holds because $x \geq 0$ and $y < 0$, implying $|x| \leq |x - y|$.

In all cases, $|\text{ReLU}(x) - \text{ReLU}(y)| \leq |x - y|$. Therefore, ReLU is **1-Lipschitz continuous**.

### B.3 SIGMOID FUNCTION

The Sigmoid activation function is defined as:

$$\sigma(x) = \frac{1}{1 + e^{-x}}.$$

The derivative of the Sigmoid function is:

$$\sigma'(x) = \sigma(x)(1 - \sigma(x)).$$

Using the fact that $\forall x \in \mathbb{R}$, $\sigma(x) > 0$, $1 - \sigma(x) > 0$, we apply AM-GM inequality:

$$\frac{\sigma(x) + (1 - \sigma(x))}{2} = \frac{1}{2} \geq \sqrt{\sigma(x)(1 - \sigma(x))}.$$

Squaring both sides,

$$\left(\frac{1}{2}\right)^2 = \frac{1}{4} \geq \sigma(x)(1 - \sigma(x)).$$

Thus,

$$0 \leq \sigma'(x) = \sigma(x)(1 - \sigma(x)) \leq \frac{1}{4}.$$

By the Mean Value Theorem, for any $x, y \in \mathbb{R}$, there exists some $c$ between $x$ and $y$ such that:

$$|\sigma(x) - \sigma(y)| = |\sigma'(c)||x - y|.$$

Using that $|\sigma'(c)| \leq \frac{1}{4}$, we have for all $x, y \in \mathbb{R}$

$$|\sigma(x) - \sigma(y)| \leq \frac{1}{4}|x - y|.$$

Therfore, Sigmoid is **$\frac{1}{4}$-Lipschitz continuous**.

### B.4 GAUSSIAN ERROR LINEAR UNIT(GELU)

The GELU activation function is expressed as:

$$\text{GELU}(x) = x\Phi(x),$$

where $\Phi(x)$ is the cumulative distribution function (CDF) of the standard normal distribution:

$$\Phi(x) = \frac{1}{2}\left(1 + \text{erf}\left(\frac{x}{\sqrt{2}}\right)\right).$$

First, we compute the derivative of $\text{GELU}(x)$:

$$\frac{d}{dx}\text{GELU}(x) = \Phi(x) + x\phi(x),$$

where $\phi(x)$ is the probability density function (PDF) of the standard normal distribution:

$$\phi(x) = \frac{1}{\sqrt{2\pi}}e^{-x^2/2}.$$

In order to show the boundedness of the derivative, we examine the second derivative:

$$\frac{d^2}{dx^2}\text{GELU}(x) = 2\phi(x) - x^2\phi(x).$$

Setting the second derivative equal to zero to find critical points:

$$\frac{d^2}{dx^2}\text{GELU}(x) = 0 \implies \phi(x)(2 - x^2) = 0.$$

Since $\phi(x) > 0$ for all $x \in \mathbb{R}$, the extrema of $\frac{d}{dx}\text{GELU}(x)$ occurs at $x = \pm\sqrt{2}$.

Hence, it is enough to examine the value of $\frac{d}{dx}\text{GELU}(x) = \Phi(x) + x\phi(x)$ at $\pm\infty, \pm\sqrt{2}$ :

$$\lim_{x\to\infty} \frac{d}{dx}\text{GELU}(x) = \lim_{x\to\infty} \Phi(x) + \lim_{x\to\infty} x\phi(x) = 1 + 0 = 1$$

$$\lim_{x\to-\infty} \frac{d}{dx}\text{GELU}(x) = \lim_{x\to-\infty} \Phi(x) + \lim_{x\to-\infty} x\phi(x) = 0 + 0 = 0$$

$$\frac{d}{dx}\text{GELU}(\sqrt{2}) = \Phi(\sqrt{2}) + \sqrt{2}\cdot\phi(\sqrt{2}) = \frac{1}{2}(1 + \text{erf}(1)) + \sqrt{2}\frac{1}{\sqrt{2\pi}}e^{-1} \approx 1.129$$

$$\frac{d}{dx}\text{GELU}(-\sqrt{2}) = \Phi(-\sqrt{2}) - \sqrt{2}\cdot\phi(-\sqrt{2}) = \frac{1}{2}(1 + \text{erf}(-1)) - \sqrt{2}\frac{1}{\sqrt{2\pi}}e^{-1} \approx -0.129$$

using that

$$\lim_{x\to\infty} \Phi(x) = 1, \lim_{x\to-\infty} \Phi(x) = 0, \lim_{x\to\pm\infty} x\phi(x) = \lim_{x\to\pm\infty} \frac{1}{\sqrt{2\pi}} x e^{-x^2/2} = 0$$

Thus,

$$\left|\frac{d}{dx}\text{GELU}(x)\right| \le 1.13, \quad \forall x \in \mathbb{R}.$$

Therefore, GELU is **1.13-Lipschitz continuous**.

## C Pseudoinverse and Gradient-based Training

**Performance Comparison Between Pseudoinverse and Training.** While we used the pseudoinverse to compute the weight matrix in the ER layer, as shown in Equation (5), it is also possible to obtain this weight matrix through gradient-based training. For this comparison, we set up a mean squared error (MSE) loss between $\boldsymbol{H}_{\mathcal{N}}^{(l)} - \mathbb{E}_{\phi}[\tilde{\boldsymbol{H}}_{\mathcal{N}}^{(l)}]$ and $\boldsymbol{H}^{(l-1)}\boldsymbol{W}^{(l)}$, and optimized it using the Adam optimizer with a learning rate of 0.01, applying early stopping with a patience of 10 epochs.

As shown in Table 5, the pseudoinverse generally provided better performance, demonstrating its effectiveness compared to training. However, training also produced comparable results in most cases. Since computation time varies by dataset, as discussed in the next paragraph, we suggest that training can be considered a viable alternative to the pseudoinverse within the FILLER framework.

**Time consumption comparison between using pseudoinverse and training.** We measured the average time consumption of the pseudoinverse computation during a single iteration of our proposed method, FILLER. Since the pseudoinverse is computed separately at each layer, the values reported represent the sum over all $L$ layers of the GNN. We also measured the time consumption of gradient-based training in a single iteration as an alternative of pseudoinverse. The detailed time measurements are presented in Table 6.

Table 5: Performance comparison between post-processed models using either the pseudoinverse or gradient-based training to obtain the weight matrix in FILLER. Bold numbers indicate higher accuracy, and colored values highlight statistically significant differences at the 5% level ($p < 0.05$). The pseudoinverse outperforms training in 25 out of 30 cases.

| | -50% | | | -75% | | | -100% | | |
|---|---|---|---|---|---|---|---|---|---|
| **Dataset** | P-inverse | Train | Δ | P-inverse | Train | Δ | P-inverse | Train | Δ |
| Cora | $\mathbf{76.9 \pm 2.0}$ | $76.8 \pm 2.0$ | $\downarrow 0.1\%$ | $\mathbf{74.1 \pm 2.3}$ | $73.8 \pm 2.3$ | $\downarrow 0.4\%$ | $\mathbf{74.6 \pm 2.1}$ | $74.1 \pm 2.0$ | $\downarrow 0.7\%$ |
| Citeseer | $\mathbf{68.5 \pm 2.0}$ | $68.3 \pm 2.1$ | $\downarrow 0.2\%$ | $\mathbf{68.5 \pm 2.2}$ | $68.5 \pm 2.3$ | $\downarrow 0.0\%$ | $\mathbf{69.4 \pm 2.5}$ | $69.3 \pm 2.6$ | $\downarrow 0.1\%$ |
| Pubmed | $\mathbf{72.2 \pm 4.2}$ | $71.9 \pm 4.4$ | $\downarrow 0.5\%$ | $\mathbf{71.7 \pm 5.6}$ | $71.4 \pm 5.9$ | $\downarrow 0.5\%$ | $\mathbf{73.6 \pm 3.9}$ | $72.7 \pm 3.7$ | $\downarrow 1.3\%$ |
| Computers | $\mathbf{79.8 \pm 3.3}$ | $79.8 \pm 3.3$ | $\downarrow 0.0\%$ | $\mathbf{77.2 \pm 3.2}$ | $77.2 \pm 3.2$ | $\downarrow 0.0\%$ | $\mathbf{69.2 \pm 4.8}$ | $68.0 \pm 6.1$ | $\downarrow 1.7\%$ |
| Photo | $\mathbf{89.4 \pm 2.7}$ | $89.4 \pm 2.7$ | $\downarrow 0.0\%$ | $\mathbf{87.8 \pm 2.8}$ | $87.8 \pm 2.9$ | $\downarrow 0.1\%$ | $\mathbf{85.4 \pm 2.4}$ | $84.5 \pm 2.3$ | $\downarrow 1.0\%$ |
| CS | $89.4 \pm 0.5$ | $\mathbf{89.4 \pm 0.5}$ | $\uparrow 0.0\%$ | $88.2 \pm 0.5$ | $\mathbf{88.2 \pm 0.5}$ | $\uparrow 0.0\%$ | $90.6 \pm 0.5$ | $\mathbf{90.6 \pm 0.5}$ | $\uparrow 0.0\%$ |
| Physics | $\mathbf{91.3 \pm 1.4}$ | $91.2 \pm 1.4$ | $\downarrow 0.0\%$ | $90.1 \pm 1.6$ | $\mathbf{90.1 \pm 1.6}$ | $\uparrow 0.0\%$ | $\mathbf{90.4 \pm 1.6}$ | $90.3 \pm 1.8$ | $\downarrow 0.1\%$ |
| Flickr | $\mathbf{50.1 \pm 0.2}$ | $50.1 \pm 0.2$ | $\downarrow 0.0\%$ | $46.4 \pm 0.5$ | $\mathbf{46.4 \pm 0.6}$ | $\uparrow 0.0\%$ | $\mathbf{46.2 \pm 0.3}$ | $46.2 \pm 0.3$ | $\downarrow 0.0\%$ |
| Ogbn-arxiv | $\mathbf{67.1 \pm 0.2}$ | $67.1 \pm 0.2$ | $\downarrow 0.0\%$ | $\mathbf{62.3 \pm 0.1}$ | $62.2 \pm 0.1$ | $\downarrow 0.1\%$ | $\mathbf{49.8 \pm 1.0}$ | $48.6 \pm 1.1$ | $\downarrow 2.5\%$ |
| Reddit | $\mathbf{95.7 \pm 0.0}$ | $95.7 \pm 0.0$ | $\downarrow 0.0\%$ | $\mathbf{94.9 \pm 0.1}$ | $94.9 \pm 0.1$ | $\downarrow 0.0\%$ | $\mathbf{55.9 \pm 1.2}$ | $54.9 \pm 1.6$ | $\downarrow 1.7\%$ |

The time complexity of the pseudoinverse is quadratic in the feature dimension $d$ and linear in the number of nodes $n$. Consequently, its computational cost is significantly influenced by the feature size of the graph dataset. As shown in Table 6, the pseudoinverse takes longer than training when the feature dimension is relatively large while faster when feature dimension is small. Conversely, when the graph size increases but the feature dimension is small, training time increases while the pseudoinverse computation becomes relatively faster due to the reduced feature size. Graph datasets usually utilize node features processed from raw data (Hou et al., 2023), such as bag-of-words representations in Cora or average word embeddings from skip-gram in OGBN-Arxiv (Hu et al., 2020), resulting in typically small feature dimensions. Therefore, the pseudoinverse computation is a realistic and even scalable solution for large-scale graphs datasets.

However, for graphs with a large number of features (e.g., raw features), the computational cost of the pseudoinverse becomes prohibitive. In such situations, alternative methods can be employed, such as using gradient-based training instead of the pseudoinverse or applying dimensionality reduction techniques like PCA as a pre-processing of dataset. As described in previous paragraph, the pseudoinverse provides better and more stable results in our post-processing framework. Therefore, whenever feasible, we recommend using the pseudoinverse computation for improved performance.

Table 6: Dataset statistics and time consumption for pseudoinverse and gradient-based training in a single iteration of FILLER. Time is measured in seconds ($s$).

| | Cora | Citeser | Pubemd | Computers | Photo | CS | Physics | Flickr | Ogbn-arxiv | Reddit |
|---|---|---|---|---|---|---|---|---|---|---|
| P-inv | 0.176 | 0.477 | 0.039 | **0.072** | **0.059** | 14.992 | 55.382 | **0.051** | **0.041** | **0.086** |
| Train | **0.074** | **0.079** | **0.038** | 0.085 | 0.083 | **0.253** | **0.471** | 0.135 | 0.175 | 0.269 |
| # nodes | 2,708 | 3,327 | 19,717 | 13,752 | 7,650 | 18,333 | 34,493 | 89,250 | 169,343 | 232,965 |
| # edges | 10,556 | 9,104 | 88,648 | 491,722 | 238,162 | 163,788 | 495,924 | 899,756 | 1,166,243 | 114,615,892 |
| # features | 1,433 | 3,703 | 500 | 767 | 745 | 6,805 | 8,415 | 500 | 128 | 602 |
| # classes | 7 | 6 | 3 | 10 | 8 | 15 | 5 | 7 | 40 | 41 |

## D Condition for convergence

**Lemma D.1.** *The following process-wise condition guarantees that both C2 and C3 hold:*

$$\forall l,\ \forall t, \quad \frac{\|\boldsymbol{\epsilon}^{(l)}(t+1, t, \tilde{\boldsymbol{A}}, \tilde{\boldsymbol{A}})\|_{\mathrm{F}}}{\|\boldsymbol{\epsilon}^{(l)}(t+1, t, \boldsymbol{A}, \boldsymbol{A})\|_{\mathrm{F}}} \geq 1 + \alpha,$$

*where $\tilde{\boldsymbol{A}}$ is a matrix obtained by masking some of the elements of $\boldsymbol{A}$ to zero, $\alpha > 0$ is any fixed constant, and $\|\cdot\|_{\mathrm{F}}$ represents the Frobenius norm.*

*Proof.* Assume that the above condition holds. By triangle inequality,

$$\|\boldsymbol{\epsilon}^{(l)}(t+1, t+1, \boldsymbol{A}, \tilde{\boldsymbol{A}})\|_{\mathrm{F}} \leq \|\boldsymbol{\epsilon}^{(l)}(t+1, t, \boldsymbol{A}, \boldsymbol{A})\|_{\mathrm{F}} + \|\boldsymbol{\epsilon}^{(l)}(t, t+1, \boldsymbol{A}, \tilde{\boldsymbol{A}})\|_{\mathrm{F}}. \quad \cdots (*)$$

Here, note that

$$\|\boldsymbol{\epsilon}^{(l)}(t, t+1, \boldsymbol{A}, \tilde{\boldsymbol{A}})\|_{\mathrm{F}} = (1-k)\|\boldsymbol{\epsilon}^{(l)}(t, t, \boldsymbol{A}, \tilde{\boldsymbol{A}})\|_{\mathrm{F}}$$

holds by our construction Eq(1) and collinearity of $\boldsymbol{\epsilon}^{(l)}(t, t+1, \boldsymbol{A}, \tilde{\boldsymbol{A}})$ and $\boldsymbol{\epsilon}^{(l)}(t, t, \boldsymbol{A}, \tilde{\boldsymbol{A}})$.

Also, by the condition,

$$\|\boldsymbol{\epsilon}^{(l)}(t+1, t, \boldsymbol{A}, \boldsymbol{A})\|_{\mathrm{F}} \leq \frac{1}{1+\alpha}\|\boldsymbol{\epsilon}^{(l)}(t+1, t, \tilde{\boldsymbol{A}}, \tilde{\boldsymbol{A}})\|_{\mathrm{F}} = \frac{1}{1+\alpha}\, k\|\boldsymbol{\epsilon}^{(l)}(t, t, \boldsymbol{A}, \tilde{\boldsymbol{A}})\|_{\mathrm{F}}.$$

Combining all together, $*$ can be extended as follows:

$$\delta_{t+1}^{(l)} = \|\boldsymbol{\epsilon}^{(l)}(t+1, t+1, \boldsymbol{A}, \tilde{\boldsymbol{A}})\|_{\mathrm{F}} \leq \left(1 - \frac{\alpha}{1+\alpha}k\right)\|\boldsymbol{\epsilon}^{(l)}(t, t, \boldsymbol{A}, \tilde{\boldsymbol{A}})\|_{\mathrm{F}} = \left(1 - \frac{\alpha}{1+\alpha}k\right)\delta_t^{(l)}$$

This demonstrates that after every iteration, the gap between $\mathrm{AGG}_t^{(l)}(\boldsymbol{H}^{(l-1)}, \boldsymbol{A})$ and $\mathrm{AGG}_t^{(l)}(\boldsymbol{H}^{(l-1)}, \tilde{\boldsymbol{A}})$ is bounded by $1 - \frac{\alpha}{1+\alpha}k\,(<1)$ ratio of the previous term w.r.t. $t$.

Recursively applying this inequality leads to

$$\delta_t^{(l)} \leq \left(1 - \frac{\alpha}{1+\alpha}k\right)^t \delta_0^{(l)}$$

where convergence can be proved after taking $t \to \infty$

$$0 \leq \delta_\infty^{(l)} \leq \lim_{t\to\infty}\left(1 - \frac{\alpha}{1+\alpha}k\right)^t \delta_0^{(l)} = 0$$

□

**Theorem D.2.** *Our proposed $h_D$, the simplest and intuitive form of $h$, satisfies the all three conditions C1, C2, and C3.*

*Proof.* Put $h = h_D$ where $h_D$ is constructed as follows:

$$h_D(\boldsymbol{H}^{(l-1)}\boldsymbol{W}^{(l)}, \boldsymbol{A}) = \left(\frac{|V|}{\sum_{i=1}^{|V|}\sum_{j=1}^{|V|}\hat{A}_{i,j}}\right)^n \cdot \boldsymbol{H}^{(l-1)}\boldsymbol{W}^{(l)}$$

(1) (Condition C1) Let $\boldsymbol{A} = \boldsymbol{O}$, then $\hat{\boldsymbol{A}} = \boldsymbol{A} + \boldsymbol{I}_{|V|} = \boldsymbol{I}_{|V|}$.

$$h_D(\boldsymbol{H}^{(l-1)}\boldsymbol{W}^{(l)}, \boldsymbol{O}) = \left(\frac{|V|}{\sum_{i=1}^{|V|}\sum_{j=1}^{|V|}(\boldsymbol{I}_{|V|})_{i,j}}\right)^n \cdot \boldsymbol{H}^{(l-1)}\boldsymbol{W}^{(l)} = \boldsymbol{H}^{(l-1)}\boldsymbol{W}^{(l)}$$

(2) (Condition C2, C3) It suffices to show that $h_D$ satisfies the condition in Lemma 3.1.

$$\frac{\|\boldsymbol{\epsilon}^{(l)}(t+1, t, \tilde{\boldsymbol{A}}, \tilde{\boldsymbol{A}})\|_{\mathrm{F}}}{\|\boldsymbol{\epsilon}^{(l)}(t+1, t, \boldsymbol{A}, \boldsymbol{A})\|_{\mathrm{F}}} = \frac{\|k h_D(\boldsymbol{H}^{(l-1)}\boldsymbol{W}^{(l)}, \tilde{\boldsymbol{A}})\|_{\mathrm{F}}}{\|k h_D(\boldsymbol{H}^{(l-1)}\boldsymbol{W}^{(l)}, \boldsymbol{A})\|_{\mathrm{F}}}$$

$$= \left\| \frac{k\left(\frac{|V|}{\sum_{i=1}^{|V|}\sum_{j=1}^{|V|}\hat{\tilde{A}}_{i,j}}\right)^n \cdot \boldsymbol{H}^{(l-1)}\boldsymbol{W}^{(l)}}{k\left(\frac{|V|}{\sum_{i=1}^{|V|}\sum_{j=1}^{|V|}\hat{A}_{i,j}}\right)^n \cdot \boldsymbol{H}^{(l-1)}\boldsymbol{W}^{(l)}} \right\|_{\mathrm{F}}$$

$$= \left( \frac{\sum_{i=1}^{|V|} \sum_{j=1}^{|V|} \hat{A}_{i,j}}{\sum_{i=1}^{|V|} \sum_{j=1}^{|V|} \hat{\tilde{A}}_{i,j}} \right)^n = \left( \frac{|V| + 2|E|}{|V| + 2|\tilde{E}|} \right)^n$$

where $\tilde{E}$ denotes the edge set of the graph associated with $\tilde{A}$.

Note that $\tilde{E}$ contains strictly fewer edges than $E$, i.e. $|\tilde{E}| \leq |E| - 1$.

Thus,

$$\left( \frac{|V| + 2|E|}{|V| + 2|\tilde{E}|} \right)^n \geq \left( \frac{|V| + 2|E|}{|V| + 2(|E| - 1)} \right)^n = \left( 1 + \frac{2}{|V| + 2|E| - 2} \right)^n$$

and $\alpha$ can be set to $\dfrac{2}{|V| + 2|E| - 2}$ to satisfy the condition of the Lemma 3.1.

□

**Remark.** *In large graph datasets, the $\alpha$ mentioned in the above proposition tends to be very small. However, this is solely because we have set $\alpha$ to be a small value intentionally for the purpose of mathematical proof. In practice, when comparing with a graph where edges are perturbed to some extent, $\alpha$ can be set to a reasonably large value, resulting in faster convergence rates.*

*For example, suppose that edges are removed by 50% in Reddit dataset as in our experiment. In such case,*

$$\left( \frac{|V| + 2|E|}{|V| + 2|\tilde{E}|} \right) = \left( \frac{232965 + 114615892}{232965 + 57307946} \right) \geq 1.99$$

*Then we can set $\alpha = 0.99$ assuming that $n = 1$, $k = 0.2$.*

*By the proof of the theorem, we have*

$$\frac{\|\boldsymbol{\epsilon}^{(l)}(5, 5, \boldsymbol{A}, \tilde{\boldsymbol{A}})\|_{\mathrm{F}}}{\|\boldsymbol{\epsilon}^{(l)}(0, 0, \boldsymbol{A}, \tilde{\boldsymbol{A}})\|_{\mathrm{F}}} \leq \left( 1 - \frac{0.99}{1 + 0.99} \cdot 0.2 \right)^5 \simeq \mathbf{0.59}$$

*which implies the gap between the original aggregated representation and perturbed one has reduced by more than 40% compared to the initial point.*

*Similarly, if edges are removed by 75%,*

$$\left( \frac{|V| + 2|E|}{|V| + 2|\tilde{E}|} \right) = \left( \frac{232965 + 114615892}{232965 + 28653973} \right) \geq 3.97.$$

*Here, we can set $\alpha = 2.97$ which results to*

$$\frac{\|\boldsymbol{\epsilon}^{(l)}(5, 5, \boldsymbol{A}, \tilde{\boldsymbol{A}})\|_{\mathrm{F}}}{\|\boldsymbol{\epsilon}^{(l)}(0, 0, \boldsymbol{A}, \tilde{\boldsymbol{A}})\|_{\mathrm{F}}} \leq \left( 1 - \frac{2.97}{1 + 2.97} \cdot 0.2 \right)^5 \simeq \mathbf{0.44}.$$

*Finally, for the complete absence of the edges, we get*

$$\left( \frac{|V| + 2|E|}{|V| + 2|\tilde{E}|} \right) = \left( \frac{232965 + 114615892}{232965} \right) \geq 492.$$

*Setting $\alpha = 491$ leads to*

$$\frac{\|\boldsymbol{\epsilon}^{(l)}(5,5,\boldsymbol{A},\tilde{\boldsymbol{A}})\|_{\mathrm{F}}}{\|\boldsymbol{\epsilon}^{(l)}(0,0,\boldsymbol{A},\tilde{\boldsymbol{A}})\|_{\mathrm{F}}} \leq \left(1-\frac{491}{1+491}\cdot 0.2\right)^5 \simeq \mathbf{0.33}.$$

*These computational results demonstrate that the GAP can be effectively bridged even with minimal post-processing and small $k$. Furthermore, in graphs with fully removed edges, merely five processing steps suffice to reduce the gap by less than one-third.*

## E Architecture Details

We used five different architectures in our experiments: SAGE, GCN, SGC, GAT, and GIN. The AGG and UPDATE functions are defined differently depending on the specific architecture. Even within the same architecture, there can be multiple ways to define these operations. Therefore, we provide the exact definitions of AGG and UPDATE used for each architecture in our experiments. Additionally, for larger datasets, we employed the inductive variants of GCN and SGC, as their original forms are not designed for neighbor sampling.

**Graph Convolutional Network (Kipf & Welling, 2016)** GCN performs a linear transformation followed by normalized aggregation of node features and the node features are updated with a non-linear activation:

$$\boldsymbol{H}_{\mathcal{N}}^{(l)} = \hat{\boldsymbol{D}}^{-1/2}\hat{\boldsymbol{A}}\hat{\boldsymbol{D}}^{-1/2}\boldsymbol{H}^{(l-1)}\boldsymbol{W}^{(l)}, \qquad \boldsymbol{H}^{(l)} = \sigma(\boldsymbol{H}_{\mathcal{N}}^{(l)}),$$

where $\hat{\boldsymbol{A}} = \boldsymbol{A} + \boldsymbol{I}$ is the adjacency matrix with self-loops, and $\hat{\boldsymbol{D}}$ is the corresponding degree matrix. $\boldsymbol{W}^{(l)}$ is the weight matrix for the linear transformation at layer $l$. In the inductive version, normalization can be formulated as:

$$\boldsymbol{H}_{\mathcal{N}}^{(l)} = \hat{\boldsymbol{D}}^{-1}\hat{\boldsymbol{A}}\boldsymbol{H}^{(l-1)}\boldsymbol{W}^{(l)}.$$

**GraphSAGE (Hamilton et al., 2017)** GraphSAGE aggregates information from neighbors using various strategies (e.g., mean, LSTM, or max). In our case, we use the mean aggregation version:

$$\boldsymbol{H}_{\mathcal{N}}^{(l)} = \boldsymbol{M}\boldsymbol{A}\boldsymbol{H}^{(l-1)}\boldsymbol{W}^{(l)} + \boldsymbol{H}^{(l-1)}\boldsymbol{B}^{(l)}, \qquad \boldsymbol{H}^{(l)} = \sigma(\boldsymbol{H}_{\mathcal{N}}^{(l)})$$

where the diagonal matrix $\boldsymbol{M}$ normalizes based on node degrees, with diagonal elements defined as:

$$\boldsymbol{M}_{ii} = \begin{cases} \frac{1}{|\mathcal{N}(i)|} & \text{if } |\mathcal{N}(i)| > 0 \\ 0 & \text{if node i is isolated (i.e., no neighbors).} \end{cases}$$

Here, $\boldsymbol{B}^{(l)}$ is an additional weight matrix applied to the node's own features, allowing the model to combine both the aggregated neighbor information and the node's previous representation. The final node embeddings, $\boldsymbol{H}^{(l)}$, are then obtained by applying a nonlinearity $\sigma$ to the combined result.

**Simplified Graph Convolution (Wu et al., 2019)** SGC simplifies GCN by removing the non-linear activation functions, making the UPDATE operation an identity function. While SGC can be viewed as a single-layer model performing $k$-hop aggregation in one AGG operation, we treat each 1-hop aggregation as a separate layer's AGG operation.

This incremental approach allows the ER to recover 1-hop aggregated representations step by step, instead of recovering the entire $k$ -hop aggregation at once. The formulation for SGC is as follows, with the linear transformation applied only at the first layer:

- First layer:

$$\boldsymbol{H}_{\mathcal{N}}^{(1)} = \hat{\boldsymbol{D}}^{-1/2}\hat{\boldsymbol{A}}\hat{\boldsymbol{D}}^{-1/2}\boldsymbol{X}\boldsymbol{W}^{(1)}, \qquad \boldsymbol{H}^{(l)} = \boldsymbol{H}_{\mathcal{N}}^{(l)}$$

- Subsequent layers (without transformation):

$$\boldsymbol{H}_{\mathcal{N}}^{(l)} = \hat{\boldsymbol{D}}^{-1/2}\hat{\boldsymbol{A}}\hat{\boldsymbol{D}}^{-1/2}\boldsymbol{H}^{(l-1)} \quad \text{for } l \geq 2, \qquad \boldsymbol{H}^{(l)} = \boldsymbol{H}_{\mathcal{N}}^{(l)}$$

In the inductive version, normalization is handled similarly to GCN, as SGC can be considered a linear variant of GCN.

**Graph Attention Network (Veličković et al., 2017)** GAT introduces learnable attention weights for each edge, enabling the model to weigh the importance of neighboring nodes during aggregation. The aggregation is formulated as:

$$\boldsymbol{H}_{\mathcal{N}}^{(l)} = \boldsymbol{A}^{*(l)} \boldsymbol{H}_j^{(l-1)} \boldsymbol{W}^{(l)}, \qquad \boldsymbol{H}^{(l)} = \sigma(\boldsymbol{H}_{\mathcal{N}}^{(l)})$$

Where element of $\boldsymbol{A}^{*(l)}$, $\alpha_{ij}^{(l)}$ is the attention coefficient for edge (i,j) , computed as:

$$\alpha_{ij}^{(l)} = \frac{\exp\left(\text{LeakyReLU}\left(\mathbf{a}^\top [\boldsymbol{W}^{(l)} \mathbf{h}_i^{(l-1)} \| \boldsymbol{W}^{(l)} \mathbf{h}_j^{(l-1)}]\right)\right)}{\sum_{k \in \mathcal{N}(i) \cup \{i\}} \exp\left(\text{LeakyReLU}\left(\mathbf{a}^\top [\boldsymbol{W}^{(l)} \mathbf{h}_i^{(l-1)} \| \boldsymbol{W}^{(l)} \mathbf{h}_k^{(l-1)}]\right)\right)}$$

where $\mathbf{a}$ is a learnable vector, and $\|$ denotes concatenation. Through this attention mechanism, GAT allows each node to focus more on specific neighbors, enhancing the aggregation process based on learned importance.

**Graph Isomorphism Network (Xu et al., 2018)** GIN aggregates information from neighbors through summation and applies a multi-layer perceptron (MLP) for transformation, allowing it to better distinguish between different graph structures:

$$\boldsymbol{H}_{\mathcal{N}}^{(l)} = \text{MLP}^{(l)}(\boldsymbol{A}\boldsymbol{H}^{(l-1)} + (1 + \epsilon^{(l)})\boldsymbol{H}^{(l-1)}), \qquad \boldsymbol{H}^{(l)} = \sigma(\boldsymbol{H}_{\mathcal{N}}^{(l)}),$$

where $\epsilon^{(l)}$ is a learnable scalar or a fixed constant; in our experiments, we used a learnable scalar.

GIN's use of summation for neighbor aggregation leads to substantial differences in representation when training with neighbor sampling versus using the full edge information during inference. As a result, we excluded GIN from experiments on large-scale datasets.

As shown in Appendix G, FILLER enhances the robustness of GIN, despite its unique aggregation (summation) and transformation (MLP) processes compared to other architectures. Notably, even though GIN's transformation layer is more expressive than the ER layer, ER is still able to mitigate representation shifts to some extent. Interestingly, although the base GIN model initially showed lower performance compared to other architectures, as it is not specifically designed for node classification tasks, FILLER significantly enhanced its performance.

# F Experiment Setting Details

We evaluated our method, FILLER, using seven widely adopted small-scale node classification datasets and three large-scale datasets. The small-scale datasets, based on prior work (Shchur et al., 2018), include three citation networks (Cora, Citeseer, and Pubmed), two co-purchase graphs (Amazon-Computers and Amazon-Photo), and two co-authorship networks (Coauthor-CS and Coauthor-Physics). Following Shchur et al. (2018), we only considered the largest connected component of each graph. For the citation networks, we used 20 nodes per class for training, 500 nodes for validation, and 1000 nodes for testing. For the other datasets, we allocated 20 nodes per class for training, 30 nodes per class for validation, and the remaining nodes for testing.

In the large-scale experiments, we used the Flickr dataset (Zeng et al., 2019), the Reddit dataset (Hamilton et al., 2017), and the OGBN-Arxiv dataset (Hu et al., 2020), following the fixed public splits provided in the original papers.

The primary objective of our experiments was to assess whether FILLER enhances model robustness without sacrificing adaptability, rather than optimizing individual model performance. Therefore, we aimed to maintain consistent hyperparameters across datasets and architectures, as shown in Table 7. For large-scale datasets, we used a separate set of hyperparameters with deeper and wider architectures to account for the increased complexity of the relationships between inputs and labels in larger graphs.

Some architecture-specific adjustments were made: (1) GAT: Due to its multi-head architecture, we used 8 heads, which means hidden size is divided into 8 heads. The dropout probability for attention coefficients was set to 0.3. (2) SGC: We did not apply dropout to the features, as the original paper (Wu et al., 2019) does not use dropout for the node classification task. Additionally,

| Dataset | *Small* | *Large* |
|---|---|---|
| # layers | 2 | 3 |
| hidden dim | 64 | 256 |
| activation | relu | relu |
| learning rate | 0.01 | 0.003 |
| weight decay | 0.001 | 0 |
| dropout | 0.5 | 0.5 |
| max epochs | 1000 | 200 |
| patience | 100 | 50 |
| fan out | - | [15, 10, 5] |

Table 7: Hyperparameter settings used for small and large datasets across different architectures.

no activation function was used, since SGC is a linear model. (3) GIN: A 2-layer MLP was used with the hidden dimension set to $2 \times d_l$ at layer $l$. Since GIN is not tailored for node classification, it exhibited unstable performance and could not be trained on the co-purchase datasets under the same settings. As a result, we adjusted the learning rate to 0.003 and applied feature normalization by row-normalizing the features on the co-purchase datasets which ensure attributes summed to one.

For batch training, we employed the neighbor sampler (Hamilton et al., 2017) with a batch size of 1024. The model was trained in batches, while all edges were used during inference.

In the edge removal and restoration experiments, edges were treated as undirected. We used *random.sample* to randomly select edges for removal or restoration. Additionally, the sequence of edge removal and restoration was randomized independently for each run using a different random seed.

## G Full Experiment Results

In the following two pages, we present the complete results of our edge removal (Table 8) and edge restoration (Table 9) experiments across all datasets and architectures.

**Edge Removal Experiments.** For small-scale datasets, FILLER improves performance across all architectures by an average of 1.35%, 4.01%, and 21.75% when 50%, 75%, and 100% of the edges are removed, respectively. In large-scale datasets, FILLER delivers average improvements across all architectures of 0.33%, 2.56%, and 28.49% under the same edge removal conditions. These consistent improvements across all datasets and architectures highlight FILLER 's effectiveness as a general post-processing technique for GNNs.

Although performance gains are relatively smaller in larger datasets when edges are partially removed(50%, 75%), FILLER shows substantial improvements when all edges are removed. This can be explained by the use of neighbor sampling during training on large datasets, which allows models to encounter graph perturbations during training. As a result, models experience less severe performance degradation when moderate sparsification occurs. However, even with neighbor sampling, models struggle to maintain robustness in highly sparsified graphs, leading to more significant performance declines when many edges are removed. In such cases, FILLER greatly improves robustness, demonstrating its effectiveness on batch-trained models in large datasets.

**Edge Restoration Experiments.** In the edge restoration experiments, GNNs were trained on 60% of the edges and tested with 0%, 50%, and 100% of the excluded edges restored. The post-processed models shows improved accuracy as more informative edges were restored, just as the base models did, demonstrating that FILLER preserves the edge-adaptability of GNNs. During edge restoration, the post-processed models do not show performance drop compared to the base model in 127 out of 141 cases at the 1% significance level.

Most of the accuracy drops (14 cases) occurred in large-scale datasets, while accuracy improvements were seen in small datasets. We hypothesize that this discrepancy is due to the use of neighbor sampling during training on large-scale datasets. Since neighbor sampling exposes the model to a limited number of edges during training, the model already experience adaptation to additional edges during inference, since all edges are used in inference; even before they are explicitly restored in testing. In

contrast, small-scale datasets, which rely on full-batch training, show a clearer relationship between edge restoration and performance gains.

This suggests that edge restoration in batch-trained models may require a different level of edge adaptability compared to models trained with full-batch methods. Further investigation into this distinction is left for future work.

Table 8: Accuracy after applying FILLER with 50%, 75%, and 100% removal of edges. The numbers are in bold and colored in the same manner as in Table 1. FILLER consistently and significantly improves the accuracy of the base model across all architectures, demonstrating its effectiveness and versatility in enhancing the robustness of existing GNNs.

| | -50% | | | -75% | | | -100% | | |
|---|---|---|---|---|---|---|---|---|---|
| **Dataset** | SAGE | + FILLER | Δ | SAGE | + FILLER | Δ | SAGE | + FILLER | Δ |
| Cora | 74.7 ± 2.6 | **76.9 ± 2.0** | ↑ 2.9% | 68.7 ± 3.5 | **74.1 ± 2.3** | ↑ 7.8% | 57.9 ± 4.9 | **74.6 ± 2.1** | ↑ 28.9% |
| Citeseer | 66.5 ± 2.4 | **68.5 ± 2.0** | ↑ 3.0% | 62.7 ± 2.7 | **68.5 ± 2.2** | ↑ 9.3% | 56.4 ± 4.5 | **69.4 ± 2.5** | ↑ 23.1% |
| Pubmed | 71.9 ± 4.0 | 72.2 ± 4.2 | ↑ 0.4% | 69.5 ± 5.5 | **71.7 ± 5.6** | ↑ 3.2% | 65.4 ± 8.8 | **73.6 ± 3.9** | ↑ 12.5% |
| Computers | 79.6 ± 3.3 | 79.8 ± 3.3 | ↑ 0.2% | 76.3 ± 3.6 | **77.2 ± 3.2** | ↑ 1.2% | 43.5 ± 10.1 | **69.2 ± 4.8** | ↑ 58.9% |
| Photo | 89.4 ± 2.8 | 89.4 ± 2.7 | ↑ 0.0% | 87.3 ± 2.8 | **87.8 ± 2.8** | ↑ 0.6% | 67.8 ± 7.5 | **85.4 ± 2.4** | ↑ 25.9% |
| CS | 88.8 ± 0.5 | **89.4 ± 0.5** | ↑ 0.6% | 86.5 ± 0.8 | **88.2 ± 0.5** | ↑ 2.0% | 82.8 ± 2.4 | **90.6 ± 0.5** | ↑ 9.3% |
| Physics | 91.0 ± 1.5 | **91.3 ± 1.4** | ↑ 0.3% | 89.0 ± 2.0 | **90.1 ± 1.6** | ↑ 1.2% | 80.6 ± 5.7 | **90.4 ± 1.6** | ↑ 12.1% |
| Flickr | 50.2 ± 0.2 | 50.1 ± 0.2 | ↓ 0.4% | 46.2 ± 0.6 | 46.4 ± 0.5 | ↑ 0.5% | 41.9 ± 0.5 | **46.2 ± 0.3** | ↑ 10.2% |
| Ogbn-arxiv | 67.2 ± 0.2 | 67.1 ± 0.2 | ↓ 0.1% | 61.7 ± 0.3 | **62.3 ± 0.1** | ↑ 0.9% | 44.3 ± 1.1 | **49.8 ± 1.0** | ↑ 12.6% |
| Reddit | 95.7 ± 0.0 | 95.7 ± 0.0 | ↑ 0.0% | 94.9 ± 0.1 | 94.9 ± 0.1 | ↑ 0.0% | 45.9 ± 1.1 | **55.9 ± 1.2** | ↑ 21.9% |

| | -50% | | | -75% | | | -100% | | |
|---|---|---|---|---|---|---|---|---|---|
| **Dataset** | GCN | + FILLER | Δ | GCN | + FILLER | Δ | GCN | + FILLER | Δ |
| Cora | 77.8 ± 1.8 | **79.1 ± 1.0** | ↑ 1.7% | 74.2 ± 2.1 | **76.7 ± 1.7** | ↑ 3.5% | 69.2 ± 2.9 | **77.4 ± 1.6** | ↑ 11.9% |
| Citeseer | 68.7 ± 2.4 | **70.1 ± 2.4** | ↑ 2.0% | 66.7 ± 3.0 | **69.9 ± 2.2** | ↑ 4.8% | 64.2 ± 3.0 | **70.3 ± 2.2** | ↑ 9.5% |
| Pubmed | 76.4 ± 1.7 | 76.5 ± 1.9 | ↑ 0.1% | 75.5 ± 1.9 | **76.2 ± 2.1** | ↑ 1.0% | 73.6 ± 2.2 | **76.6 ± 2.1** | ↑ 4.0% |
| Computers | 82.6 ± 2.2 | 82.7 ± 2.3 | ↑ 0.1% | 81.1 ± 2.1 | **81.4 ± 2.1** | ↑ 0.3% | 69.2 ± 2.1 | **75.8 ± 2.8** | ↑ 9.6% |
| Photo | 90.6 ± 1.5 | 90.6 ± 1.5 | ↑ 0.0% | 89.5 ± 1.6 | 89.7 ± 1.5 | ↑ 0.1% | 82.6 ± 2.8 | **87.1 ± 2.0** | ↑ 5.4% |
| CS | 89.5 ± 0.3 | **89.7 ± 0.3** | ↑ 0.2% | 87.9 ± 0.4 | **88.6 ± 0.3** | ↑ 0.7% | 87.8 ± 0.8 | **91.6 ± 0.3** | ↑ 4.4% |
| Physics | 91.8 ± 1.0 | **92.0 ± 0.9** | ↑ 0.2% | 90.5 ± 1.2 | **91.1 ± 1.1** | ↑ 0.6% | 87.8 ± 2.0 | **92.0 ± 1.0** | ↑ 4.8% |
| Flickr | 47.0 ± 0.5 | **47.8 ± 0.3** | ↑ 1.7% | 38.8 ± 1.0 | **42.3 ± 0.6** | ↑ 8.8% | 27.7 ± 1.2 | **43.6 ± 0.5** | ↑ 57.4% |
| Ogbn-arxiv | 67.8 ± 0.1 | 67.8 ± 0.1 | ↑ 0.0% | 62.4 ± 0.2 | **63.1 ± 0.2** | ↑ 1.1% | 42.2 ± 0.6 | **48.8 ± 0.8** | ↑ 15.8% |
| Reddit | 93.6 ± 0.1 | 93.6 ± 0.0 | ↑ 0.0% | 92.9 ± 0.0 | **92.9 ± 0.0** | ↑ 0.0% | 43.4 ± 0.5 | 44.9 ± 1.3 | ↑ 3.3% |

| | -50% | | | -75% | | | -100% | | |
|---|---|---|---|---|---|---|---|---|---|
| **Dataset** | SGC | + FILLER | Δ | SGC | + FILLER | Δ | SGC | + FILLER | Δ |
| Cora | 77.2 ± 1.9 | **78.3 ± 1.6** | ↑ 1.5% | 73.5 ± 1.4 | **76.4 ± 1.8** | ↑ 3.9% | 68.3 ± 1.9 | **77.0 ± 1.7** | ↑ 12.8% |
| Citeseer | 67.2 ± 1.8 | **68.2 ± 2.0** | ↑ 1.6% | 64.9 ± 2.6 | **67.9 ± 2.5** | ↑ 4.6% | 62.1 ± 2.6 | **68.4 ± 1.8** | ↑ 10.1% |
| Pubmed | 74.9 ± 2.2 | 74.9 ± 2.2 | ↑ 0.0% | 74.3 ± 2.1 | 74.6 ± 1.9 | ↑ 0.3% | 73.2 ± 1.9 | **74.6 ± 2.3** | ↑ 1.9% |
| Computers | 82.5 ± 1.5 | 82.6 ± 1.5 | ↑ 0.1% | 81.4 ± 1.4 | 81.5 ± 1.5 | ↑ 0.2% | 72.5 ± 2.5 | **78.0 ± 1.3** | ↑ 7.6% |
| Photo | 90.4 ± 1.9 | 90.5 ± 2.0 | ↑ 0.1% | 89.4 ± 2.1 | 89.6 ± 1.9 | ↑ 0.3% | 83.7 ± 2.7 | **87.8 ± 1.9** | ↑ 4.8% |
| CS | 89.4 ± 0.6 | **89.6 ± 0.7** | ↑ 0.2% | 87.8 ± 0.5 | **88.5 ± 0.5** | ↑ 0.9% | 87.7 ± 0.7 | **91.5 ± 0.6** | ↑ 4.3% |
| Physics | 91.9 ± 1.3 | **92.0 ± 1.3** | ↑ 0.2% | 90.8 ± 1.4 | **91.2 ± 1.2** | ↑ 0.5% | 88.6 ± 1.7 | **92.0 ± 1.4** | ↑ 3.8% |
| Flickr | 46.1 ± 0.2 | **47.3 ± 0.1** | ↑ 2.6% | 37.3 ± 0.9 | **41.6 ± 0.6** | ↑ 11.5% | 25.8 ± 0.7 | **44.3 ± 0.3** | ↑ 71.9% |
| Ogbn-arxiv | 65.8 ± 0.1 | **65.9 ± 0.2** | ↑ 0.1% | 60.9 ± 0.2 | **61.7 ± 0.1** | ↑ 1.2% | 41.5 ± 0.1 | **49.8 ± 0.1** | ↑ 19.9% |
| Reddit | 93.6 ± 0.1 | 93.6 ± 0.1 | ↑ 0.0% | 92.6 ± 0.0 | 92.7 ± 0.1 | ↑ 0.0% | 29.6 ± 1.6 | **51.1 ± 1.1** | ↑ 72.5% |

| | -50% | | | -75% | | | -100% | | |
|---|---|---|---|---|---|---|---|---|---|
| **Dataset** | GAT | + FILLER | Δ | GAT | + FILLER | Δ | GAT | + FILLER | Δ |
| Cora | 77.7 ± 1.8 | **79.1 ± 1.6** | ↑ 1.8% | 74.5 ± 1.7 | **78.4 ± 2.0** | ↑ 5.1% | 69.9 ± 1.8 | **80.2 ± 1.6** | ↑ 14.8% |
| Citeseer | 69.2 ± 1.6 | 69.7 ± 1.9 | ↑ 0.8% | 67.2 ± 1.7 | **69.9 ± 1.7** | ↑ 4.1% | 65.6 ± 1.8 | **70.5 ± 1.6** | ↑ 7.6% |
| Pubmed | 75.8 ± 2.3 | **76.9 ± 2.2** | ↑ 1.4% | 74.8 ± 2.4 | **77.1 ± 2.4** | ↑ 3.1% | 73.3 ± 2.7 | **77.8 ± 2.3** | ↑ 6.1% |
| Computers | 83.1 ± 2.2 | **83.3 ± 2.1** | ↑ 0.3% | 80.9 ± 2.5 | 81.1 ± 2.3 | ↑ 0.3% | 65.3 ± 5.5 | **78.7 ± 2.8** | ↑ 20.6% |
| Photo | 91.1 ± 1.4 | 91.0 ± 1.3 | ↓ 0.0% | 89.8 ± 1.8 | 89.8 ± 1.4 | ↓ 0.1% | 80.7 ± 3.2 | **88.9 ± 1.8** | ↑ 10.2% |
| CS | 88.3 ± 0.4 | **88.6 ± 0.4** | ↑ 0.3% | 87.2 ± 0.4 | **87.8 ± 0.4** | ↑ 0.8% | 87.6 ± 0.8 | **89.4 ± 0.5** | ↑ 2.1% |
| Physics | 90.7 ± 1.7 | **91.0 ± 1.6** | ↑ 0.3% | 89.4 ± 2.1 | **90.2 ± 1.7** | ↑ 0.9% | 86.5 ± 3.2 | **90.6 ± 1.4** | ↑ 4.7% |
| Flickr | 49.5 ± 0.2 | 49.5 ± 0.2 | ↓ 0.0% | 42.3 ± 0.4 | **44.7 ± 0.3** | ↑ 5.8% | 29.5 ± 1.4 | **42.3 ± 0.0** | ↑ 43.6% |
| Ogbn-arxiv | 68.1 ± 0.2 | 68.1 ± 0.2 | ↑ 0.0% | 63.2 ± 0.1 | **63.8 ± 0.2** | ↑ 1.0% | 44.9 ± 0.2 | **51.4 ± 0.8** | ↑ 14.3% |
| Reddit | 93.7 ± 0.1 | 93.7 ± 0.1 | ↑ 0.0% | 93.1 ± 0.0 | 93.1 ± 0.1 | ↑ 0.0% | 51.9 ± 0.5 | 51.1 ± 0.5 | ↓ 1.5% |

| | -50% | | | -75% | | | -100% | | |
|---|---|---|---|---|---|---|---|---|---|
| **Dataset** | GIN | + FILLER | Δ | GIN | + FILLER | Δ | GIN | + FILLER | Δ |
| Cora | 71.6 ± 2.8 | **73.6 ± 1.4** | ↑ 2.8% | 63.1 ± 4.4 | **69.9 ± 2.6** | ↑ 10.7% | 49.2 ± 9.9 | **61.8 ± 4.2** | ↑ 25.7% |
| Citeseer | 61.7 ± 5.3 | **65.2 ± 3.0** | ↑ 5.7% | 56.9 ± 8.9 | **64.8 ± 3.4** | ↑ 14.0% | 47.9 ± 14.9 | **64.2 ± 4.2** | ↑ 34.0% |
| Pubmed | 69.3 ± 4.6 | **74.8 ± 3.3** | ↑ 7.9% | 65.4 ± 6.8 | **73.5 ± 4.0** | ↑ 12.4% | 56.2 ± 11.2 | **70.2 ± 4.9** | ↑ 25.0% |
| Computers | 71.7 ± 4.3 | **72.8 ± 4.2** | ↑ 1.5% | 57.6 ± 7.7 | **63.5 ± 6.5** | ↑ 10.4% | 21.4 ± 10.5 | **49.2 ± 4.1** | ↑ 129.6% |
| Photo | 82.2 ± 3.4 | **84.3 ± 3.2** | ↑ 2.5% | 69.9 ± 4.7 | **79.0 ± 3.6** | ↑ 13.0% | 24.8 ± 8.2 | **56.3 ± 4.2** | ↑ 127.0% |
| CS | 81.1 ± 1.5 | **84.8 ± 1.1** | ↑ 4.6% | 74.8 ± 3.0 | **84.2 ± 1.1** | ↑ 12.6% | 55.9 ± 10.1 | **80.9 ± 1.2** | ↑ 44.8% |
| Physics | 87.5 ± 2.4 | **89.2 ± 1.6** | ↑ 1.9% | 82.7 ± 4.0 | **87.8 ± 2.0** | ↑ 6.1% | 56.8 ± 17.2 | **81.5 ± 4.1** | ↑ 43.6% |

Table 9: Accuracy change after applying FILLER with 0%, 50%, and 100% restoration of missing edges. The numbers are in bold and colored in the same way as in Table 1. FILLER preserves the adaptability of GNNs to informative edges while enhancing robustness across all architectures, demonstrating its stability and versatility for GNNs.

| | 0% | | | +50% | | | +100% | | |
|---|---|---|---|---|---|---|---|---|---|
| **Dataset** | SAGE | + FILLER | Δ | SAGE | + FILLER | Δ | SAGE | + FILLER | Δ |
| Cora | $74.2 \pm 2.3$ | $\mathbf{75.1 \pm 2.0}$ | $\uparrow$ 1.3% | $77.0 \pm 2.0$ | $\mathbf{77.6 \pm 2.1}$ | $\uparrow$ 0.8% | $79.1 \pm 1.9$ | $\mathbf{79.4 \pm 2.0}$ | $\uparrow$ 0.4% |
| Citeseer | $66.0 \pm 3.1$ | $\mathbf{66.9 \pm 3.3}$ | $\uparrow$ 1.4% | $68.6 \pm 2.8$ | $68.6 \pm 2.9$ | $\uparrow$ 0.0% | $69.6 \pm 2.3$ | $\mathbf{70.0 \pm 2.3}$ | $\uparrow$ 0.7% |
| Pubmed | $72.6 \pm 2.6$ | $72.8 \pm 3.0$ | $\uparrow$ 0.3% | $73.6 \pm 2.8$ | $73.4 \pm 2.8$ | $\downarrow$ 0.3% | $74.6 \pm 2.7$ | $74.1 \pm 2.9$ | $\downarrow$ 0.6% |
| Computers | $78.4 \pm 2.9$ | $78.5 \pm 2.9$ | $\uparrow$ 0.1% | $79.1 \pm 3.1$ | $79.1 \pm 3.0$ | $\uparrow$ 0.1% | $79.6 \pm 3.1$ | $79.6 \pm 3.1$ | $\uparrow$ 0.0% |
| Photo | $90.3 \pm 2.3$ | $\mathbf{90.4 \pm 2.3}$ | $\uparrow$ 0.1% | $90.8 \pm 2.2$ | $90.9 \pm 2.2$ | $\uparrow$ 0.1% | $91.1 \pm 2.1$ | $91.1 \pm 2.1$ | $\uparrow$ 0.0% |
| CS | $89.2 \pm 0.4$ | $\mathbf{89.7 \pm 0.4}$ | $\uparrow$ 0.5% | $90.4 \pm 0.5$ | $\mathbf{90.5 \pm 0.5}$ | $\uparrow$ 0.1% | $91.1 \pm 0.5$ | $\mathbf{91.2 \pm 0.5}$ | $\uparrow$ 0.1% |
| Physics | $91.2 \pm 0.8$ | $\mathbf{91.3 \pm 0.9}$ | $\uparrow$ 0.2% | $91.7 \pm 0.9$ | $91.8 \pm 1.0$ | $\uparrow$ 0.1% | $92.1 \pm 1.0$ | $92.1 \pm 1.0$ | $\uparrow$ 0.1% |
| Flickr | $50.9 \pm 0.2$ | $50.9 \pm 0.1$ | $\downarrow$ 0.0% | $\mathbf{51.6 \pm 0.2}$ | $51.5 \pm 0.2$ | $\downarrow$ 0.3% | $52.0 \pm 0.3$ | $51.9 \pm 0.3$ | $\downarrow$ 0.1% |
| Ogbn-arxiv | $69.0 \pm 0.0$ | $68.9 \pm 0.1$ | $\downarrow$ 0.1% | $\mathbf{70.2 \pm 0.1}$ | $70.1 \pm 0.0$ | $\downarrow$ 0.2% | $\mathbf{71.0 \pm 0.2}$ | $71.0 \pm 0.1$ | $\downarrow$ 0.1% |
| Reddit | $95.9 \pm 0.1$ | $95.9 \pm 0.1$ | $\downarrow$ 0.0% | $96.1 \pm 0.0$ | $\mathbf{96.1 \pm 0.0}$ | $\uparrow$ 0.0% | $96.3 \pm 0.0$ | $96.3 \pm 0.0$ | $\downarrow$ 0.0% |

| | 0% | | | +50% | | | +100% | | |
|---|---|---|---|---|---|---|---|---|---|
| **Dataset** | GCN | + FILLER | Δ | GCN | + FILLER | Δ | GCN | + FILLER | Δ |
| Cora | $77.6 \pm 1.5$ | $\mathbf{78.2 \pm 1.7}$ | $\uparrow$ 0.8% | $80.1 \pm 1.7$ | $80.4 \pm 1.6$ | $\uparrow$ 0.4% | $81.7 \pm 1.5$ | $\mathbf{82.1 \pm 1.4}$ | $\uparrow$ 0.5% |
| Citeseer | $68.5 \pm 2.0$ | $\mathbf{68.9 \pm 2.0}$ | $\uparrow$ 0.6% | $69.9 \pm 2.2$ | $69.9 \pm 2.0$ | $\uparrow$ 0.0% | $70.7 \pm 2.0$ | $\mathbf{70.9 \pm 2.1}$ | $\uparrow$ 0.3% |
| Pubmed | $75.8 \pm 1.9$ | $75.9 \pm 1.9$ | $\uparrow$ 0.1% | $76.6 \pm 1.7$ | $76.5 \pm 1.9$ | $\downarrow$ 0.1% | $77.0 \pm 1.7$ | $76.9 \pm 1.7$ | $\downarrow$ 0.1% |
| Computers | $83.0 \pm 1.5$ | $83.0 \pm 1.5$ | $\uparrow$ 0.0% | $83.3 \pm 1.5$ | $83.4 \pm 1.5$ | $\uparrow$ 0.1% | $83.7 \pm 1.6$ | $83.6 \pm 1.5$ | $\downarrow$ 0.0% |
| Photo | $90.7 \pm 1.6$ | $90.7 \pm 1.7$ | $\uparrow$ 0.0% | $91.0 \pm 1.7$ | $91.0 \pm 1.6$ | $\uparrow$ 0.0% | $91.0 \pm 1.7$ | $91.0 \pm 1.6$ | $\uparrow$ 0.0% |
| CS | $89.9 \pm 0.5$ | $\mathbf{90.1 \pm 0.5}$ | $\uparrow$ 0.2% | $90.6 \pm 0.5$ | $90.7 \pm 0.5$ | $\uparrow$ 0.1% | $91.1 \pm 0.4$ | $\mathbf{91.1 \pm 0.4}$ | $\uparrow$ 0.1% |
| Physics | $92.2 \pm 0.7$ | $\mathbf{92.4 \pm 0.7}$ | $\uparrow$ 0.2% | $92.6 \pm 0.7$ | $\mathbf{92.7 \pm 0.7}$ | $\uparrow$ 0.1% | $92.9 \pm 0.7$ | $\mathbf{93.0 \pm 0.7}$ | $\uparrow$ 0.1% |
| Flickr | $\mathbf{49.5 \pm 0.1}$ | $49.2 \pm 0.2$ | $\downarrow$ 0.6% | $\mathbf{50.5 \pm 0.2}$ | $50.1 \pm 0.3$ | $\downarrow$ 1.0% | $\mathbf{51.0 \pm 0.3}$ | $50.4 \pm 0.4$ | $\downarrow$ 1.2% |
| Ogbn-arxiv | $\mathbf{68.8 \pm 0.1}$ | $68.7 \pm 0.1$ | $\downarrow$ 0.2% | $\mathbf{70.0 \pm 0.1}$ | $69.8 \pm 0.2$ | $\downarrow$ 0.2% | $\mathbf{70.7 \pm 0.1}$ | $70.6 \pm 0.2$ | $\downarrow$ 0.2% |
| Reddit | $93.7 \pm 0.1$ | $93.7 \pm 0.0$ | $\downarrow$ 0.0% | $93.9 \pm 0.0$ | $93.9 \pm 0.0$ | $\uparrow$ 0.0% | $\mathbf{93.9 \pm 0.0}$ | $93.9 \pm 0.0$ | $\downarrow$ 0.0% |

| | 0% | | | +50% | | | +100% | | |
|---|---|---|---|---|---|---|---|---|---|
| **Dataset** | SGC | + FILLER | Δ | SGC | + FILLER | Δ | SGC | + FILLER | Δ |
| Cora | $76.4 \pm 2.1$ | $\mathbf{76.9 \pm 2.2}$ | $\uparrow$ 0.7% | $79.0 \pm 1.9$ | $79.3 \pm 2.0$ | $\uparrow$ 0.4% | $80.9 \pm 1.8$ | $\mathbf{81.4 \pm 1.7}$ | $\uparrow$ 0.6% |
| Citeseer | $67.1 \pm 3.1$ | $67.3 \pm 3.0$ | $\uparrow$ 0.3% | $68.8 \pm 3.0$ | $68.7 \pm 2.8$ | $\downarrow$ 0.1% | $69.9 \pm 2.8$ | $70.1 \pm 2.9$ | $\uparrow$ 0.3% |
| Pubmed | $74.1 \pm 2.5$ | $\mathbf{74.3 \pm 2.4}$ | $\uparrow$ 0.4% | $74.6 \pm 2.6$ | $74.8 \pm 2.6$ | $\uparrow$ 0.3% | $74.8 \pm 2.7$ | $\mathbf{75.0 \pm 2.7}$ | $\uparrow$ 0.3% |
| Computers | $82.7 \pm 1.4$ | $82.7 \pm 1.4$ | $\uparrow$ 0.0% | $82.9 \pm 1.5$ | $83.0 \pm 1.4$ | $\uparrow$ 0.1% | $83.1 \pm 1.4$ | $83.1 \pm 1.4$ | $\uparrow$ 0.0% |
| Photo | $90.3 \pm 2.2$ | $90.3 \pm 2.3$ | $\uparrow$ 0.0% | $90.5 \pm 2.2$ | $90.5 \pm 2.3$ | $\uparrow$ 0.0% | $90.7 \pm 2.3$ | $90.7 \pm 2.3$ | $\uparrow$ 0.0% |
| CS | $89.9 \pm 0.8$ | $\mathbf{90.0 \pm 0.7}$ | $\uparrow$ 0.1% | $90.6 \pm 0.8$ | $90.7 \pm 0.8$ | $\uparrow$ 0.0% | $91.1 \pm 0.8$ | $\mathbf{91.1 \pm 0.8}$ | $\uparrow$ 0.0% |
| Physics | $92.3 \pm 1.0$ | $\mathbf{92.5 \pm 0.9}$ | $\uparrow$ 0.1% | $92.7 \pm 0.9$ | $\mathbf{92.8 \pm 0.9}$ | $\uparrow$ 0.1% | $93.0 \pm 0.9$ | $\mathbf{93.1 \pm 0.9}$ | $\uparrow$ 0.1% |
| Flickr | $\mathbf{48.9 \pm 0.2}$ | $48.5 \pm 0.1$ | $\downarrow$ 0.8% | $\mathbf{49.7 \pm 0.1}$ | $49.2 \pm 0.2$ | $\downarrow$ 1.1% | $\mathbf{50.2 \pm 0.1}$ | $49.7 \pm 0.3$ | $\downarrow$ 0.9% |
| Ogbn-arxiv | $\mathbf{66.8 \pm 0.1}$ | $66.6 \pm 0.1$ | $\downarrow$ 0.3% | $\mathbf{67.8 \pm 0.1}$ | $67.7 \pm 0.1$ | $\downarrow$ 0.3% | $\mathbf{68.5 \pm 0.1}$ | $68.3 \pm 0.1$ | $\downarrow$ 0.3% |
| Reddit | $\mathbf{93.8 \pm 0.0}$ | $93.7 \pm 0.0$ | $\downarrow$ 0.0% | $94.0 \pm 0.0$ | $94.0 \pm 0.0$ | $\uparrow$ 0.0% | $\mathbf{94.1 \pm 0.1}$ | $94.1 \pm 0.1$ | $\downarrow$ 0.0% |

| | 0% | | | +50% | | | +100% | | |
|---|---|---|---|---|---|---|---|---|---|
| **Dataset** | GAT | + FILLER | Δ | GAT | + FILLER | Δ | GAT | + FILLER | Δ |
| Cora | $77.2 \pm 1.6$ | $77.5 \pm 1.5$ | $\uparrow$ 0.4% | $79.2 \pm 1.7$ | $79.7 \pm 1.7$ | $\uparrow$ 0.6% | $81.1 \pm 1.8$ | $81.4 \pm 1.8$ | $\uparrow$ 0.4% |
| Citeseer | $69.0 \pm 2.2$ | $69.2 \pm 2.3$ | $\uparrow$ 0.2% | $70.4 \pm 2.1$ | $70.3 \pm 1.8$ | $\downarrow$ 0.2% | $71.4 \pm 1.6$ | $71.5 \pm 1.4$ | $\uparrow$ 0.2% |
| Pubmed | $75.7 \pm 1.8$ | $\mathbf{76.1 \pm 1.4}$ | $\uparrow$ 0.6% | $76.7 \pm 1.6$ | $76.7 \pm 1.3$ | $\downarrow$ 0.0% | $77.0 \pm 1.4$ | $77.2 \pm 1.2$ | $\uparrow$ 0.3% |
| Computers | $82.1 \pm 2.3$ | $82.3 \pm 2.4$ | $\uparrow$ 0.2% | $82.6 \pm 2.4$ | $82.8 \pm 2.5$ | $\uparrow$ 0.2% | $82.9 \pm 2.4$ | $82.9 \pm 2.5$ | $\uparrow$ 0.1% |
| Photo | $90.3 \pm 1.6$ | $90.2 \pm 1.7$ | $\downarrow$ 0.1% | $90.5 \pm 1.6$ | $90.4 \pm 1.8$ | $\downarrow$ 0.1% | $\mathbf{90.7 \pm 1.8}$ | $90.5 \pm 1.8$ | $\downarrow$ 0.1% |
| CS | $89.0 \pm 0.4$ | $89.0 \pm 0.4$ | $\downarrow$ 0.0% | $\mathbf{89.6 \pm 0.4}$ | $89.5 \pm 0.3$ | $\downarrow$ 0.2% | $89.9 \pm 0.4$ | $89.9 \pm 0.3$ | $\downarrow$ 0.0% |
| Physics | $91.3 \pm 1.4$ | $\mathbf{91.4 \pm 1.3}$ | $\uparrow$ 0.1% | $91.6 \pm 1.4$ | $\mathbf{91.7 \pm 1.4}$ | $\uparrow$ 0.1% | $91.8 \pm 1.4$ | $\mathbf{91.9 \pm 1.4}$ | $\uparrow$ 0.1% |
| Flickr | $\mathbf{50.9 \pm 0.1}$ | $50.3 \pm 0.2$ | $\downarrow$ 1.1% | $\mathbf{52.0 \pm 0.4}$ | $51.5 \pm 0.2$ | $\downarrow$ 0.9% | $\mathbf{52.6 \pm 0.4}$ | $52.1 \pm 0.4$ | $\downarrow$ 0.9% |
| Ogbn-arxiv | $69.1 \pm 0.2$ | $69.0 \pm 0.2$ | $\downarrow$ 0.1% | $70.3 \pm 0.1$ | $70.1 \pm 0.2$ | $\downarrow$ 0.2% | $\mathbf{71.0 \pm 0.1}$ | $70.9 \pm 0.1$ | $\downarrow$ 0.1% |
| Reddit | $\mathbf{94.0 \pm 0.1}$ | $94.0 \pm 0.1$ | $\downarrow$ 0.0% | $94.1 \pm 0.0$ | $94.1 \pm 0.1$ | $\uparrow$ 0.0% | $94.2 \pm 0.0$ | $94.2 \pm 0.0$ | $\downarrow$ 0.0% |

| | 0% | | | +50% | | | +100% | | |
|---|---|---|---|---|---|---|---|---|---|
| **Dataset** | GIN | + FILLER | Δ | GIN | + FILLER | Δ | GIN | + FILLER | Δ |
| Cora | $72.7 \pm 3.1$ | $72.8 \pm 2.5$ | $\uparrow$ 0.2% | $75.5 \pm 3.1$ | $75.9 \pm 3.3$ | $\uparrow$ 0.5% | $77.7 \pm 3.2$ | $77.7 \pm 3.2$ | $\downarrow$ 0.1% |
| Citeseer | $63.6 \pm 3.3$ | $\mathbf{64.5 \pm 2.7}$ | $\uparrow$ 1.4% | $66.4 \pm 2.8$ | $66.4 \pm 2.9$ | $\uparrow$ 0.0% | $68.3 \pm 2.7$ | $68.4 \pm 2.9$ | $\uparrow$ 0.1% |
| Pubmed | $71.8 \pm 3.0$ | $72.5 \pm 3.3$ | $\uparrow$ 0.8% | $72.7 \pm 1.9$ | $72.8 \pm 2.5$ | $\uparrow$ 0.2% | $73.6 \pm 2.1$ | $73.3 \pm 2.5$ | $\downarrow$ 0.4% |
| Computers | $76.3 \pm 2.8$ | $\mathbf{76.6 \pm 2.7}$ | $\uparrow$ 0.4% | $76.4 \pm 3.1$ | $\mathbf{76.6 \pm 3.0}$ | $\uparrow$ 0.3% | $75.9 \pm 3.4$ | $\mathbf{76.0 \pm 3.4}$ | $\uparrow$ 0.2% |
| Photo | $86.5 \pm 1.9$ | $\mathbf{86.8 \pm 2.0}$ | $\uparrow$ 0.3% | $87.1 \pm 2.1$ | $87.2 \pm 2.1$ | $\uparrow$ 0.1% | $87.0 \pm 2.5$ | $87.1 \pm 2.5$ | $\uparrow$ 0.1% |
| CS | $83.9 \pm 1.2$ | $\mathbf{84.8 \pm 1.1}$ | $\uparrow$ 1.1% | $84.8 \pm 1.1$ | $\mathbf{85.2 \pm 1.0}$ | $\uparrow$ 0.5% | $85.4 \pm 1.1$ | $\mathbf{85.7 \pm 1.0}$ | $\uparrow$ 0.3% |
| Physics | $88.8 \pm 2.1$ | $\mathbf{89.2 \pm 2.1}$ | $\uparrow$ 0.5% | $89.4 \pm 2.1$ | $\mathbf{89.6 \pm 2.1}$ | $\uparrow$ 0.3% | $89.7 \pm 2.1$ | $\mathbf{89.8 \pm 2.1}$ | $\uparrow$ 0.1% |