# OpenReview forum: "Filling in the GAP: Achieving Robust and Adaptive GNNs through Post-Processing"
_ICLR.cc/2025/Conference — ICLR 2025 Conference Withdrawn Submission_

### Official Review · Reviewer_dcqA · 2024-10-31

**Soundness:** 2
**Presentation:** 2
**Contribution:** 1
**Rating:** 3
**Confidence:** 4

**Summary:**

In this paper, the authors propose FILLER, a framework to enhance the robustness and generalization of graph machine learning against edge addition and removal through post-processing. Specifically, the proposed method mainly adopts an edge-shift recovery layer alongside the existing message-passing functions. The proposed method can be adopted as a plug-in to enhance existing GNNs. Some theoretical analyses for the proposed method are also provided. Experimental results show the effectiveness of the proposed method.

**Strengths:**

1.	Enhancing the existing GNNs to be more adaptive to structural differences between training and inference graphs is an important direction.
2.	The proposed method can be added to many different existing GNN models.
3.	Some theoretical analyses for the proposed method are provided.

**Weaknesses:**

1.	The comparison with related literature is highly insufficient. Specifically, though the authors briefly review graph distribution shift methods in Sec 4, there is no clear discussion on how the proposed method differs from the existing methods. For example, the authors argue “our work takes a different approach ... we solve the problem focusing on edge distribution shift”, but this is one most typical graph distribution shifts. Among these methods, graph test-time adaptation methods (see [1-2] and their references/citations) are particularly related, as these methods also adapt graph data to enhance the generalization ability during the inference stage.
2.	Following the above comment, in experiments, the authors only show the proposed method can enhance traditional GNNs but do not compare with any relevant literature, such as graph OOD methods. More baselines are needed to truly demonstrate the effectiveness of the proposed method over state-of-the-art baselines.
3.	Besides, since the authors also emphasize the robustness of the proposed method, more discussions and comparisons of the proposed method with existing robust graph machine learning are needed.
4.	The proposed method also seems to be related to graph structure learning literature [3] and more discussions and comparisons with the existing methods are needed.

[1] GraphTTA: Test Time Adaptation on Graph Neural Networks, ICML workshop 2022.
[2] Empowering Graph Representation Learning with Test-Time Graph Transformation, ICLR 2022.
[3] A Survey on Graph Structure Learning: Progress and Opportunities.

**Questions:**

See Weaknesses

---

### Official Review · Reviewer_buh2 · 2024-11-01

**Soundness:** 3
**Presentation:** 2
**Contribution:** 2
**Rating:** 5
**Confidence:** 2

**Summary:**

This paper considers the scenario of dynamic graphs node classification, where the graph structures can shift between training phase and test phase. This paper proposes a layer-level post-processing method to mitigate the edge distribution shift problem, thus improving the GNN robustness. Verification is both done theoretically and empirically.

**Strengths:**

- This paper provides theoretical analysis detailing how it mitigates the representation gap caused by edge distribution shifts.
- This paper gives empirical verification for most statements.
- The evaluation is across 10 graphs and 5 GNN architectures.

**Weaknesses:**

- The edge distribution shift problem is not well defined mathematically
- Lack of comparison to other robustness methods. More comprehensive comparisons should be made among other work in dynamic graph learning/GNN robustness.

**Questions:**

- The edge distribution shift can only be random removal/addition? Can it be other types of distribution shift?
- Can the proposed method be used on heterophilic graphs?

---

### Official Review · Reviewer_vMjj · 2024-11-02

**Soundness:** 3
**Presentation:** 4
**Contribution:** 2
**Rating:** 5
**Confidence:** 4

**Summary:**

The paper presents FILLER, a post-hoc method that can be applied to various GNNs to correct for edge shift, which could occur when edges are added or removed in the test graph. FILLER works by solving a so-called GNN Aggregation Perturbation (GAP) problem, that is, by trying to restore the original graph from the perturbed graph. The method is applied to several node classification tasks and different GNN architectures, where it is shown to improve (or leave unchanged) performances with respect to the vanilla counterparts.

**Strengths:**

- the paper is well written and easy to follow
- the proposed method is intuitive and simple to understand
- the evaluation is comprehensive in both depth and width (at least the part concerning its effectiveness wrt vanilla GNNs)
- thorough ablation studies are provided

**Weaknesses:**

- the paper does not investigate whether it is more effective than other methods to address edge distribution shifts (e.g. continual learning or graph domain adaptation), so it is not clear why a user should prefer it over those methods (see question 1 below).
- not clear whether there exist real-world scenarios where FILLER could actually be of use. Therefore, it is not clear whether this research will find application in real-world problems (see question 2 below).

**Questions:**

1. can you justify why a user would prefer FILLER over continual learning or graph domain adaptation strategies? Ideally, I would like to see an experimental comparison, but I understand the constraints of the rebuttal period. Even a proof-of-concept could be enough, but please bear in mind that I consider the absence of a comparison a significant limitation of this paper unless you justify this absence thorougly.
2. can you describe a real-world example where edges are added/removed from a graph at test time? To put it differently, what is a real-world use-case of FILLER? Providing an example is enough, but please do not use benchmark datasets as Cora to provide it (on benchmark datasets it is clear enough).

---

### Official Review · Reviewer_129G · 2024-11-04

**Soundness:** 3
**Presentation:** 2
**Contribution:** 3
**Rating:** 5
**Confidence:** 4

**Summary:**

This paper investigates the robustness of graph neural networks when faced with graph structure variations between the training and testing phases. It proposes FILLER, which incorporates a plug-in edge-shift recovery layer between the aggregation and update layers of GNNs. Evaluations on node classification tasks demonstrate the effectiveness of the proposed FILLER.

**Strengths:**

1）The issue of edge distribution shift, where the graph structure in the testing phase differs from that in the training phase, is a significant topic in GNNs.

2）The practice of recovering perturbed edge connections that the model may encounter during inference makes sense.

3）This paper is well-organized.

**Weaknesses:**

1）Some formulas are missing the corresponding labels, such as ER-Advanced Layer in Lines 249-251.

2) The symbols used in critical sections are challenging to discern, such as g in Equation 1, g’ in Equation 2, g_B in Equation 4, and h_D in Lines 249-251.

3）W is calculated using the matrix pseudoinverse as shown in Equation 5. Is this a contribution of this paper? If the performance enhancement is marginal and could be considered a trick, it might be best not to highlight it. Such a technique could overcomplicate the model, hindering its popularization and application.

4）A key experimental detail is missing in Figure 1: Do the results correspond to random edge deletions?

5）It is unclear the rationale behind setting $\phi(A)=O$. Does this equate to characterizing the differences between neighboring nodes and central nodes? In my opinion, it would be more beneficial to introduce random perturbations at each training epoch to bolster the robustness.

5）Why does Figure 1 indicate an enhancement in model performance when the sparsity is excessively high, for instance, beyond 0.9?

6）Does the proposed method follow the homophily assumption? If a node transformation, $g(H^{(l-1)})$, is designed to capture the differences in changes within its neighborhood, $H^{(l-1)}_N-\tilde{H}^{(l-1)}_N$, should it be assumed that the node is similar to its neighbors?

**Questions:**

Refer to Weaknesses.

---

### Note · Authors · 2024-11-14

I have read and agree with the venue's withdrawal policy on behalf of myself and my co-authors.